https://doi.org/10.1038/s41467-019-08998-1　　**OPEN**

# Hippocampal pattern separation supports reinforcement learning

Ian C. Ballard [1,2], Anthony D. Wagner [3] & Samuel M. McClure[4]

Animals rely on learned associations to make decisions. Associations can be based on relationships between object features (e.g., the three leaflets of poison ivy leaves) and outcomes (e.g., rash). More often, outcomes are linked to multidimensional states (e.g., poison ivy is green in summer but red in spring). Feature-based reinforcement learning fails when the values of individual features depend on the other features present. One solution is to assign value to multi-featural conjunctive representations. Here, we test if the hippocampus forms separable conjunctive representations that enables the learning of response contingencies for stimuli of the form: AB+, B−, AC−, C+. Pattern analyses on functional MRI data show the hippocampus forms conjunctive representations that are dissociable from feature components and that these representations, along with those of cortex, influence striatal prediction errors. Our results establish a novel role for hippocampal pattern separation and conjunctive representation in reinforcement learning.

[1] Stanford Neurosciences Graduate Training Program, Stanford University, Stanford, CA 94305, USA. [2] Helen Wills Neuroscience Institute, University of California Berkeley, Berkeley, CA 94720, USA. [3] Department of Psychology, Stanford University, Stanford, CA 94305, USA. [4] Department of Psychology, Arizona State University, Tempe, AZ 85287, USA. Correspondence and requests for materials should be addressed to I.C.B. (email: iancballard@gmail.com)

M ost North American hikers develop a reflexive aversion to poison ivy, which causes a painful rash, and learn to recognize its compound leaf with three leaflets that is green in summer and red in spring and autumn. The relationship between color and season distinguishes poison ivy from other plants like boxelder, which looks similar but is green in spring. Such learning problems are challenging because similar conjunctions of features can require different responses or elicit different predictions about future events. Responses and predictions also depend on the status of other features or context. In such problems, simple feature-response learning is insufficient and representations that include multiple features (e.g., leaf shape, color, and season) must be learned.

Learning in the brain operates over qualitatively distinct representations depending on brain system[1]. Theoretical and empirical work suggest the hippocampus rapidly forms conjunctive representations of arbitrary sets of co-occurring features[2], making the hippocampus critical for episodic memory[3]. During the encoding of conjunctive representations, hippocampal computations establish a minimal representational overlap between traces of events with partially shared features, in a process called pattern separation[4,5]. One solution to multi-featural learning problems that require stimuli with overlapping features to be associated with different outcomes is to encode neurally separable conjunctive representations, putatively through hippocampal-dependent computations, and to assign value to each pattern separated representation, putatively through hippocampal-striatal interactions. The same circuit and computational properties that make the hippocampus vital for episodic memory can also benefit striatal-dependent reinforcement learning by providing separated conjunctive representations over which value learning can occur.

Stimulus-response learning occurs by the incremental adjustment of synapses on striatal neurons[6]. Thalamic and sensory cortical inputs encode single stimuli, such as a reward-associated flash of light, and are strengthened in response to phasic reward prediction errors (PEs) encoded in dopamine release[7–9]. This system allows for incremental learning about individual feature values. Although the hippocampus is not critical for associating value with individual features or items[10], it provides dense input to the striatum[11]. Hippocampal-striatal synapses are strengthened by phasic dopamine release via D1 receptors[12] and might represent conjunctions of features distributed in space or time[6]. To test the role of the hippocampus and its interaction with the striatum in value learning over conjunctive codes, we used a non-spatial, probabilistic stimulus-response learning task including stimuli with overlapping features. We hypothesized that hippocampal pattern separation computations and hippocampal-to-striatal projections would form a conjunctive-value learning system that worked in tandem with a feature-value learning system implemented in sensory cortical-to-striatal projections.

We compared hippocampal response patterns to those of four other cortical areas that could contribute to learning in our task: perirhinal (PRc) and parahippocampal (PHc) cortices, inferior frontal sulcus (IFS), and medial orbitofrontal cortex (mOFC). The PRc and PHc gradually learn representations of individual items[13,14]. Cortical learning is generally slow to form representations linking multiple items[2], and pattern separation likely depends on hippocampal computations[5]. We therefore predicted PRc and PHc would not form pattern-separated representations of conjunctions with overlapping features. The IFS supports the representation of abstract rules[15,16] that often describe conjunctive relationships (e.g., "respond to stimuli with both features A and B"[17], but our task included design features that were intended to bias subjects away from rule-based learning. Thus we predicted the IFS would not form pattern separated

representations of conjunctions. Finally, the mOFC is involved in outcome evaluation[18–20], has been proposed to provide a state representation in learning tasks[21] and receives dense medial temporal lobe inputs[22]. Due to its prominent role in reward processing, we predicted the mOFC representations would be organized around the probability of reward associated with the stimuli, rather than stimulus features.

We designed our task and analyses to test for a hippocampal role in encoding conjunctive representations that serve as inputs for striatal associative learning. We find that the hippocampus encodes stable representations across repetitions of a stimulus, and conjunctive representations are distinct from the representations of composite features. The hippocampus also shows stronger evidence for pattern-separated conjunctive representations than PRc, PHc, IFS, and mOFC. Both the hippocampal and cortical coding are related to PE coding in the striatum. Our results suggest the hippocampus provides a pattern-separated state space that supports the learning of outcomes associated with conjunctions of sensory features.

## Results

**Behavioral results**. The subjects learned stimulus-outcome relationships that required the formation of conjunctive representations. Our task was based on the "simultaneous feature discrimination" rodent behavioral paradigm[23]. The task stimuli consisted of four feature configurations: AB, AC, B, and C. We used a speeded reaction time (RT) task in which a target "go" stimulus was differentially predicted by the four stimuli (Fig. 1). AB and C predicted the target 70% of the time and B and AC predicted the target 30% of the time. To earn money, the subjects pressed a button within a limited response window after the target onset. The response window was set adaptively for each subject and adjusted over the course of the run. Each feature was associated with the target 50% of the time, but the stimuli were more (70%) or less (30%) predictive of the target. Optimal performance required learning the value of stimuli as distinct conjunctions of features (i.e., conjunctive representations).

We first tested whether the subjects learned predictive relationships between the stimuli and the target. Subjects were faster in responding to the target when it followed stimuli that were more reliably linked to target onset (AB+ and C+) than to those that were less reliably linked (AC− and B−), $F(1,26) = 13$, $p = .001$, $\eta_G^2 = .08$, repeated measures ANOVA (Fig. 2b). Further, the reaction times (RTs) for the target-predictive stimuli decreased over the course of each run, $Z = −2.04$, $p = .041$, mixed effects model with subject as a random intercept, and the adaptive RT threshold decreased as well (Supplementary Figure 1). As a result of the faster RTs, subjects had a higher hit rate for target stimuli, $F(1,26) = 43$, $p < .001$, $\eta_G^2 = .35$, repeated measures ANOVA (Fig. 2a). In addition, subjects were more likely to make false alarms to target-predictive stimuli when the target did not appear, $F(1,26) = 34$, $p = .001$, $\eta_G^2 = .14$, repeated measures ANOVA (Fig. 2c).

We aimed to identify the mechanism by which subjects learned stimulus-outcome relationships by fitting four computational models. Note that in all models we use "value" to indicate the association between the stimulus and the target, and did not model the reward outcome of the trial.

1. No learning model: subjects ignored predictive information and responded as fast as possible after the target.
2. Feature RL: subjects learned the values for individual features but not conjunctions. For multi-featural cues, the value was updated for each feature (also called "feature weight" learning)[24].

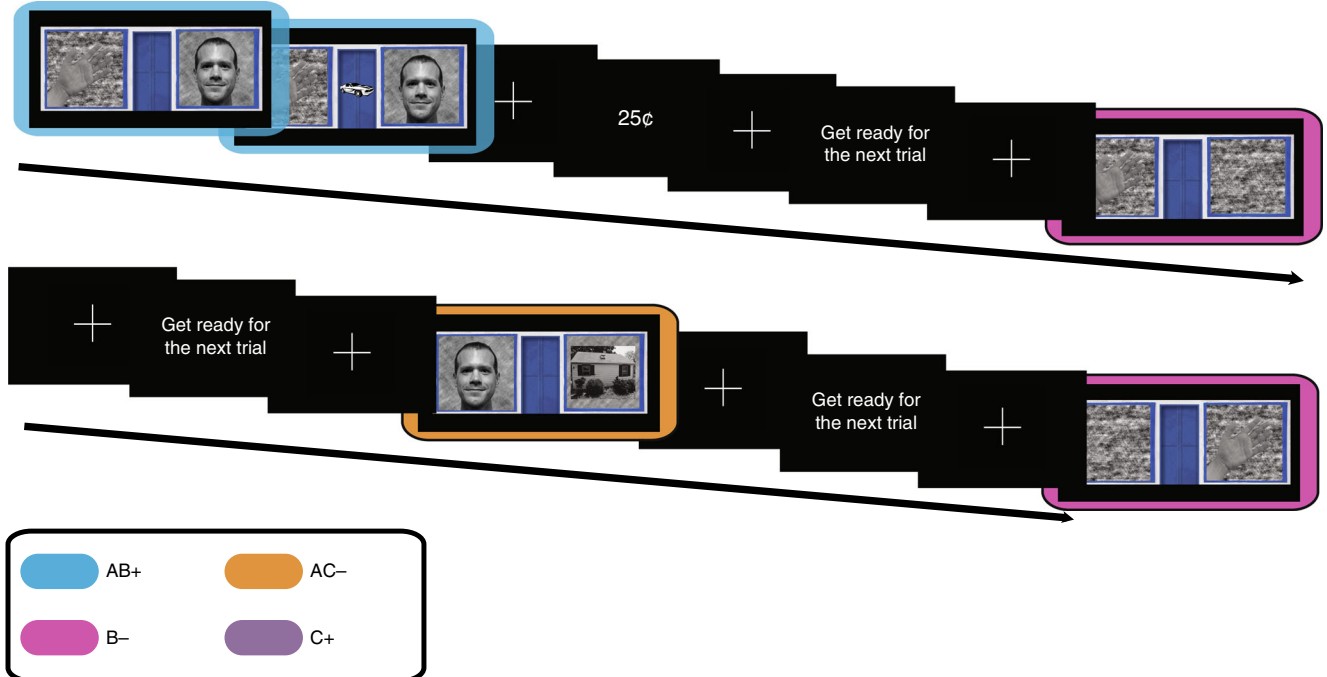

**Fig. 1** Task design. AB+, B−, and AC− trials. The target appeared at fixation 600 ms after stimulus onset. Stimuli were always presented for 2000 ms. Feedback indicated whether subjects responded quickly enough to earn a reward

3.  Conjunctive RL: subjects learned the values for each distinct stimulus. The value was updated for one representation on each trial (e.g., for "AB", the value updated for AB but not A or B).
4.  Value spread RL: subjects learned the values of stimuli but confused stimuli that shared common features (e.g., AB and B). This model spreads value updates among stimuli that share features (e.g., for AB trial, some of the value update was applied to B).

Stimuli that were highly predictive of targets were associated with faster responses, permitting us to fit each model to the RT data. We first compared the Conjunctive model with the Feature and No Learning models. Although feature learning was not adaptive for the task, it might have exerted an influence on learning[25]. Both the Feature and Conjunctive models have two free parameters: learning rate ($\alpha$) and a regression weight relating values to reaction times ($\beta$). We assessed model fits using a cross-validated predictive likelihood method. The Conjunctive model outperformed the No Learning model, $T = 136$, $p = .028$, Wilcoxon test (Fig. 2f), but was only marginally better than the Feature Model, $T = 158$, $p = .08$, Wilcoxon test, Fig. 2f. In turn, the feature model was not significantly better than No Learning, $p = .2$, Wilcoxon test. We next assessed the relative fits of these three models with a random-effects Bayesian procedure that gives the probabilities that each model would generate the data of a random subject[26]. We found the most likely model was the conjunctive model (protected exceedance probabilities (pEP): conjunctive 92.3%, feature 3.9%, no learning 3.7%, Fig. 2e). Overall, we found mixed evidence in support of learning about conjunctions.

We reasoned these results could be explained by subjects forming and learning the values of conjunctive representations while simultaneously learning some predictive values of the individual features. This behavior could arise if the hippocampal pattern separation was partially effective in encoding distinct representations for each stimulus[5] and/or the stimulus representations in the hippocampus and feature representations in cortex

were simultaneously reinforced during learning[27]. We fit a value spread model that allowed for value updates to spread between stimuli with overlapping features. A parameter $\omega$ specifies the degree to which value updates spread to other stimuli with shared features, resulting in three free parameters ($\omega, \alpha, \beta$). On the cross-validation analysis, the value spread model outperformed both the conjunctive, $T = 124$, $p = .015$, and feature models, $T = 115$, $p = .009$; Wilcoxon tests (Fig. 2f). The Bayesian model comparison confirmed the Value Spread model was the most likely model, pEP: 89.9% (Fig. 2e). Thus, although feature learning did not describe behavior better than chance, a model that incorporated a mixture of conjunctive and feature learning best described subjects behavior, Supplementary Note 1. See Supplementary Note 2 for analysis showing these effects are not due to a shift from feature to conjunctive learning over time and Supplementary Note 3 for analyses that control for effects of response rate on reaction times. The values from the value spread model were anticorrelated with reaction times, mean $r = .43$, $t(30) = 20$, $p < .001$, correlation test, Fisher corrected. In addition, the fitted regression weights for the value spread model were significantly less than 0, $T = 64$, $p < .001$, Wilcoxon test, indicating that the stimuli more strongly associated with the target were associated with faster reaction times. The fits of the spread parameter $\omega$ (mean: .44, SD: .25, Supplementary Table 1) indicated that for any given value update to the current stimulus (e.g., AB), about half that update was also applied to overlapping stimuli (e.g., B).

Our behavioral analysis showed a main effect of target association on reaction times, which is consistent with conjunctive learning and cannot be explained by feature learning (Fig. 2d). However, we observed an additional main effect of the number of features (single versus double). Subjects were faster, $F(1,26) = 7.6$, $p = .01$, $\eta_G^2 = .02$, mixed model, had a higher hit rate, $F(1,26) = 27$, $p < .001$, $\eta_G^2 = .11$, mixed model, and made more false alarms, $F(1,26) = 7.9$, $p = .008$, $\eta_G^2 = .01$, mixed model, for two feature stimuli. Feature, but not Conjunctive learning, predicted this main effect (Fig. 2d). This prediction occurred

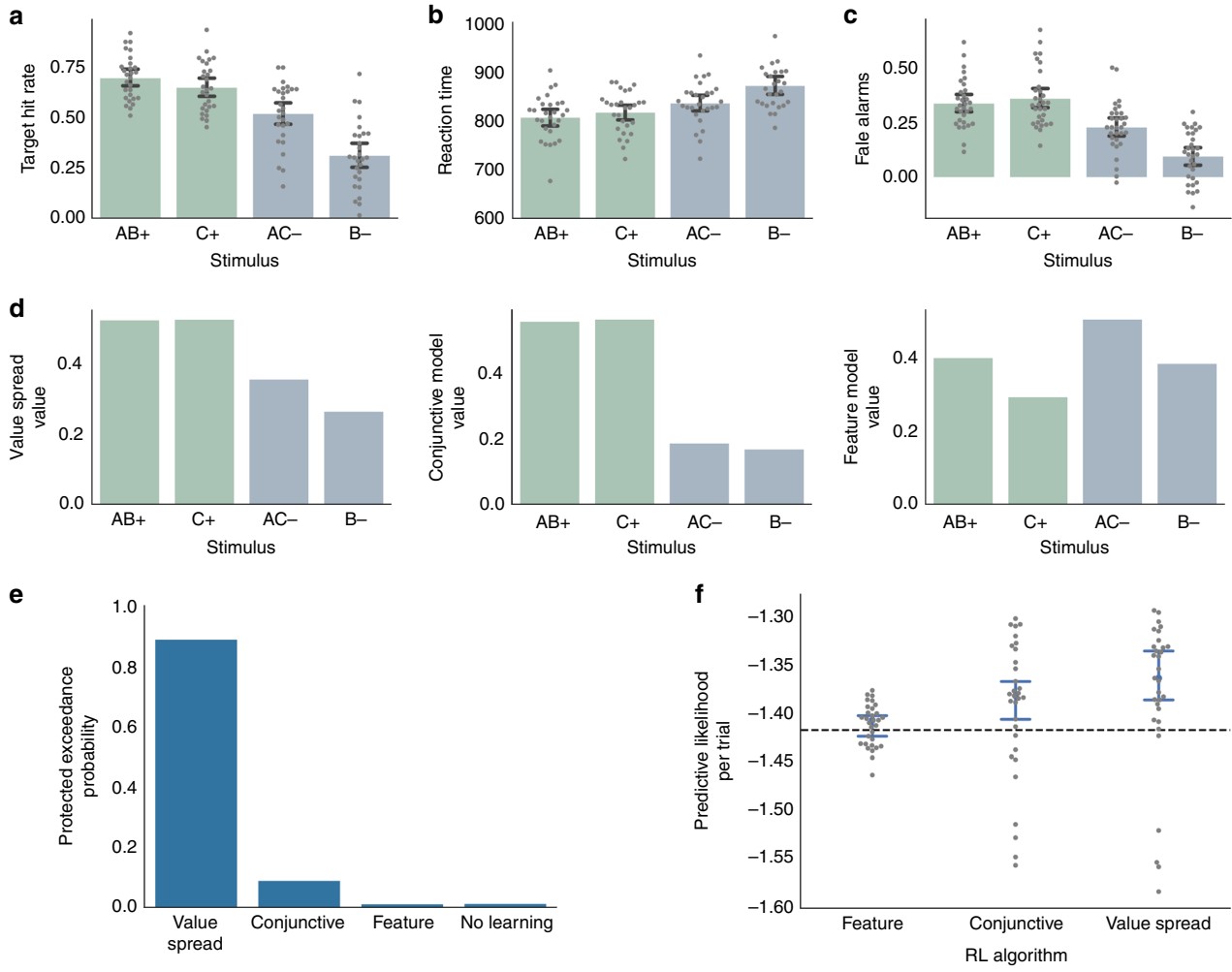

**Fig. 2** Behavior and modeling results. **a** Proportion of target trials in which the subject responded quickly enough to the target to earn a reward. Stimuli that were associated with the target (AB+, C+; green) had a higher hit rate than those that were not (AC−, B−; blue). In addition, stimuli with single features (C+, B−) were associated with a lower hit rate than those with two features (AB+, AC−). Further, this feature effect interacted with the target outcome effect. **b** Reaction times for each of the stimulus types. Reaction times were faster for stimuli associated with a target. The use of an adaptive RT threshold caused smaller differences in reaction times among conditions to translate into larger differences in hit rates in **a**. **c** False alarms for each stimulus type. Subjects were more likely to respond when no target occurred for stimuli that were associated with the target. **d** Value estimates from the value spread, conjunctive, and feature models. The conjunctive model showed an effect of target, such that AB+ and C+ had a higher value than B− and AC−. The feature model showed an effect of features, such that AB+ and AC− had a higher value than B+ and C−. Only the value spread model showed both the effect of target and the interaction between target and the number of features (present in **a–c**). **e** Bayesian random effects model comparison showed the Value Spread RL model most likely accounted for behavior. The protected exceedance probabilities summed to 1 across the models, and because they express a group random-effects measure, there are no error bars. **f** The cross-validation model comparison showed the Value Spread RL model best predicted unseen data. Log predictive likelihoods closer to 0 indicate better performance. Likelihoods are expressed per trial to normalize across differences in the number of responses among subjects. The dashed black line indicates the performance of the null model. Error bars for all panels depict bootstrapped estimates of the standard error of the group mean

because our models were initialized with zero values, which introduced a bias towards learning from the target appearance relative to target non-appearance. The bias disappears over time as values move away from zero. Because of the initial-learning bias, the A feature in the Feature model had a positive value despite being non-predictive of reward, which lead to a higher value for conjunctions. We re-fit our models with initial value as a free parameter and found that the best-fit initial value was zero for all of our models. Therefore, the behavioral performance showed signatures of both conjunctive and feature learning.

Finally, we observed an interaction among target association and the number of features, such that subjects showed a higher hit rate, $F(1,26) = 9.7$, $p = .004$, $\eta_G^2 = .05$, mixed model, and higher false alarm rate, $F(1,26) = 14.8$, $p < .001$, $\eta_G^2 = .03$, mixed model, for AC- relative to B- trials. Only the Value Spread model, which mixes Feature and Conjunctive learning, accounted for this interaction, Fig. 2d. This interaction occurred in the Value Spread model because the bias towards learning about targets rapidly overwhelms the difference in AB+ and C+ due to feature learning, whereas the relatively slower initial learning about non-targets emphasizes the difference between AC− and B−. In sum, the behavior showed patterns consistent with both conjunctive and feature learning, and the Value Spread model best accounted for the qualitative features of the data.

**Striatal prediction error analysis.** Because striatal BOLD responses track reward PEs[28], we predicted that these BOLD responses would co-vary with PEs derived from the Value Spread

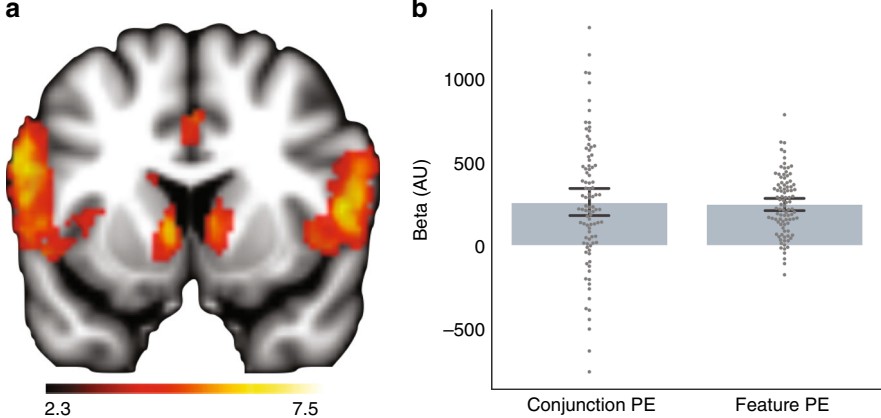

**Fig. 3** Striatal error response. **a** Regions responsive to PEs from the Feature RL model, whole-brain analysis; $p < .05$, $t$-test, FDR corrected. Color-scale refers to cluster-corrected $z$-values. **b** An ROI analysis of striatum showed that voxels with responses that scaled with PEs from the Feature RL model also scaled with PEs from the Conjunctive RL model. The feature PE bar is a statistically independent depiction of the striatal response in **a**. The conjunction PE bar shows that errors from a conjunctive learning system explained variance in striatal BOLD above and beyond errors from a feature learning system. Dots correspond to individual runs with the subject intercepts removed. Error bars depict bootstrapped estimates of the standard error of the group mean. Source data are provided as a Source Data file

model. We sought to distinguish the contribution of conjunctive learning, which learns independently about each stimulus, from that of feature learning, which causes learning to spread across stimuli that share features. Such a distinction could emerge if the striatum integrated predictions arising from inputs from feature representations in sensory cortex and conjunctive representations in the hippocampus. A feature PE regressor was constructed from the Feature RL model. A conjunctive PE regressor was constructed by computing the difference between the feature regressor and PEs computed from the Conjunctive RL model. This regressor captured unique variance associated with PEs derived from a model that learns values for conjunctions (see Methods). In addition, constructing this regressor as a difference reduced the shared variation between the feature and conjunctive regressors. However, there remained shared variation between these regressors, $r(118) = -.59$, $p < .001$, correlation test, and this shared variability reduced our ability to detect significant effects. Nonetheless, we found robust feature PE responses in the bilateral medial caudate, whole-brain corrected threshold $p < .05$; $t$-test, Fig. 3a. To confirm this finding did not occur as a result of larger responses to targets than non-targets, we extracted single trial betas from an anatomical striatal mask[29] crossed with a statistically-independent functional mask of feature PE activation. Using a mixed-effects model with random intercepts for subjects, we confirmed both the target outcome, $t(31) = 49.7$; $p < .001$; $d_z = 8.9$, and the feature PE, $t(31) = 5.7$; $p < .001$; $d_z = 1.02$, contributed to the striatal outcome response. We use $d_z$ to refer to Cohen's $d$ for paired tests. We next extracted the parameter estimates from this ROI and found these same voxels also showed evidence of a conjunction PE response, $t(31) = 4.1$; $p < .001$; $d_z = 0.72$; $t$-test, Fig. 3b, Supplementary Note 4. Note that this result was not driven by target coding because the conjunction PE difference regressor was anticorrelated with target occurrence ($r = -.28$), yet both showed a positive relationship with striatal BOLD. Together, these results confirm that striatal BOLD tracked reinforcement learning PEs that mixed learning about conjunctions and features.

**Pattern similarity analysis.** We hypothesized that the hippocampus formed conjunctive representations of task stimuli, which served as inputs to the striatum for reinforcement learning. We used a pattern similarity analysis (PSA) to probe the

representational content of the hippocampus[30]. The PSA compares the similarity of activity patterns among different trials as a function of the experimental variables of interest. We computed similarity matrices from the hippocampus, IFS, PRc, PHc, and mOFC, Supplementary Figure 4.

To be useful for learning, a region must have consistent representations across presentations of a stimulus. We ran a regression analysis on the PSA matrices to assess the similarity among the representations from different presentations of a stimulus. All ROIs had significantly higher similarity for repetitions of the same stimulus (within-stimulus similarity) than for pairs of different stimuli (between-stimulus similarity), all $p < .001$, FDR corrected, permutation test, except for the mOFC, $p > .3$. Therefore, all ROIs except for the mOFC had representations that were driven by the stimulus. Across-region comparisons showed the hippocampus had stronger within-stimulus coding than PRc, $p < .001$, PHc, $p < .001$, IFS, $p < .001$ and mOFC, $p < .001$, FDR corrected, permutation test, Fig. 4a, indicating the hippocampus had the most stable representations of the task stimuli.

Our central hypothesis was that the hippocampus, not PRc, PHc, mOFC nor IFS, would form conjunctive representations of stimuli. The representations of stimuli that shared features (AB and B) should be pattern separated, and therefore less correlated with one another, in hippocampus but should be more correlated with one another in cortical regions like PRc and PHc that provide inputs to the hippocampus. Our task intentionally included speeded responses and probabilistic outcomes, features that are known to bias subjects away from using rule-based strategies[31]. As a result, we predicted that the IFS should not have pattern-separated representations of conjunctions. Finally, because mOFC is not associated with pattern separation, we predicted that the mOFC would not have pattern separated responses. We tested whether the pattern structure in each ROI was more similar for stimuli sharing common features than for stimuli that lacked feature overlap (i.e., [(AB, AC), (AB, B), (AC, C)] versus [(AB, C), (AC, B), (B, C)]). We note that this analysis is orthogonal to the previous within-stimulus analysis and provides an independent test of stimulus coding fidelity. In addition, this regressor has only minimal covariance with the effect of response, Supplementary Note 5. We also note that all our stimuli, including single-feature stimuli, are in reality

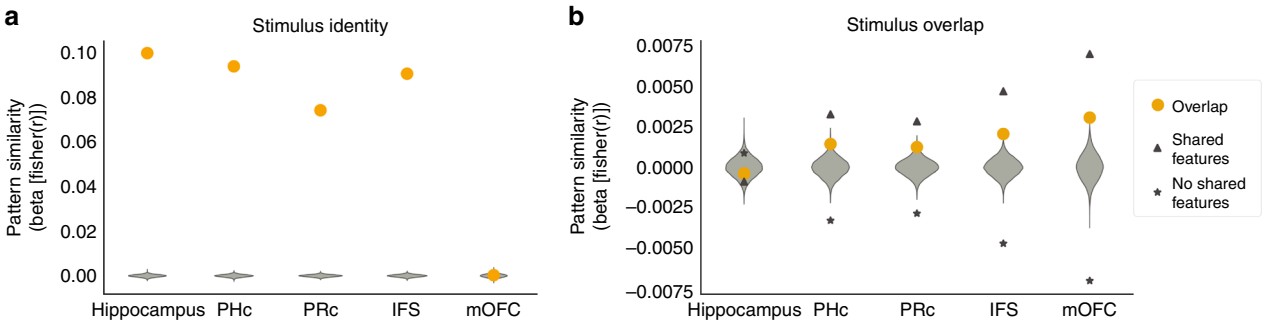

**Fig. 4** Pattern similarity analysis. **a** A regression analysis on the PSA matrices showed strong within-stimulus coding in all ROIs except mOFC, and within-stimulus coding was significantly stronger in the hippocampus relative to other regions. The *y*-axis shows the regression weights from a within-stimulus regressor on the PSA matrix of each ROI. **b** PHc, PRc, IFS, and mOFC showed increased similarity for pairs of stimuli that shared features and significantly more similarity for these pairs than the hippocampus, consistent with pattern-separated representations in the hippocampus. The triangles show an average similarity for pairs of stimuli sharing features, whereas the stars show the average similarity for pairs of stimuli not sharing features. The orange dots depict regression weights from an overlapping-versus-non-overlapping stimuli regressor on the between-stimuli correlations from the PSA matrix of each ROI. Green violins show the null distributions of regression coefficients from 10,000 randomly permuted PSA matrices. Source data are provided as a Source Data file

conjunctions of features because they are experienced in our task context (i.e., for common task context X; stimuli are truly ABX, ACX, BX, CX). Therefore, rather than testing for differences between conjunctive and non-conjunctive stimuli, our overlap regressor tested for similarities in representations between conjunctive stimuli that shared a salient feature versus those that did not. All control ROIs showed a significant effect of overlap, PRc: $p = .01$, PHc: $p = .009$, IFS: $p < .001$, and mOFC: $p < .001$, FDR corrected, permutation test, Fig. 4b, but the hippocampus did not, $p > .3$. Critically, the hippocampus showed significantly lower pattern overlap than PRc, $p = .015$, PHc, $p = .015$, IFS, $p = .002$ and mOFC, $p = .002$, FDR corrected, permutation test. Control analyses ruled out potential confounds arising from feature hemifield and reproduced these findings using a parametric mixed-effects model, Supplementary Notes 6, 7. Relative to the control ROIs, the hippocampus formed more pattern-separated conjunctive representations of stimuli.

The hippocampal representations of conjunctions could serve as inputs to the striatal reinforcement learning system. If this were the case, then variability in the formation of pattern-separated conjunctive representations in the hippocampus should correlate with striatal learning about conjunctions. To examine this relationship, we fit a mixed effects model of the conjunctive component of the striatal PE, with subject as a random intercept and hippocampal overlap as a random slope. The hippocampal overlap term was negatively related to the striatal conjunctive PE, $t(31) = -3.43$, $p = .003$, $d_z = -0.62$, Supplementary Figure 2, Supplementary Note 8. The conjunctive PE represents variance explained over-and-above the effect of feature PE and is therefore a more sensitive measure of the degree of conjunctive PE coding in the striatum. As expected, there was no relationship between hippocampal overlap and the striatal feature PE, $p > .2$, and this finding suggests that general signal quality fluctuations did not contribute to the effect. Control analyses showed this result persisted even when accounting for inter-individual and intra-individual differences in how well subjects learned (Supplementary Note 9), suggesting that it was not entirely driven by how much attention subjects paid to the task. However, attention is likely to be an important driver of both hippocampal pattern separation and striatal learning.

Given our model that representations in both sensory cortex and HPC project to the striatum to influence learning, a relative increase in overlapping representations in any of these regions

should be associated with a reduced conjunctive component of the striatal prediction error. We observed similar relationships in our medial temporal lobe (MTL) cortical ROIs and IFS, but not OFC (Supplementary Note 9). Finally, we observed a positive relationship between the strength of within-stimulus similarity in the hippocampus and striatal conjunctive PE, $t(31) = 2.49$, $p = .013$, $d_z = 0.45$, t-test, although this result depended on the exclusion of an outlier subject (Figures S2). Again, this finding suggests that the results were not driven by general signal quality issues, as stimulus identity and stimulus overlap showed opposing relationships to the conjunctive PE in the predicted direction. In sum, the more the hippocampus and medial temporal lobe cortex representations overlapped for stimuli sharing features, the less striatal error signals reflected learning signals arising from a conjunctive state space.

We next tested whether there was a relationship between hippocampal overlap and behavior. We computed an index that measured the extent to which subjects used conjunctive learning (see Methods) for each run of subjects' behavior. We fit a mixed-effects model of this measure with random intercepts for subjects and included a nuisance covariate that measured how well subjects learned in each run, relative to chance. This design helped to ensure that the relationship was not due to fluctuations in subject engagement. Contrary to our predictions, we did not find any relationship between overlap in any of our ROIs and this measure (Supplementary Note 10). We next examined whether conjunctive learning was related to univariate signal magnitude and found that runs with stronger hippocampal activity were also the runs with the most conjunctive learning, $t(31) = 3.1$, $p = .01$, $d_z = 0.56$, t-test, FDR corrected. This relationship was nonsignificant in other ROIs, all $p > 1$, t-tests.

**The effect of outcome association on stimulus representation.** How the hippocampus represents stimuli with similar associations is an open question. Computational models suggest the hippocampus supports feedback learning by representing stimuli with similar outcomes more similarly[32]. This grouping facilitates responding, while also allowing the generalization of knowledge across related items[27,33,34]. However, an alternative perspective claims that to maintain distinct representations of related items, the hippocampus orthogonalizes stimuli with related outcomes more strongly[35,36]. We conducted an exploratory analysis to assess how our ROIs represented stimuli according to the

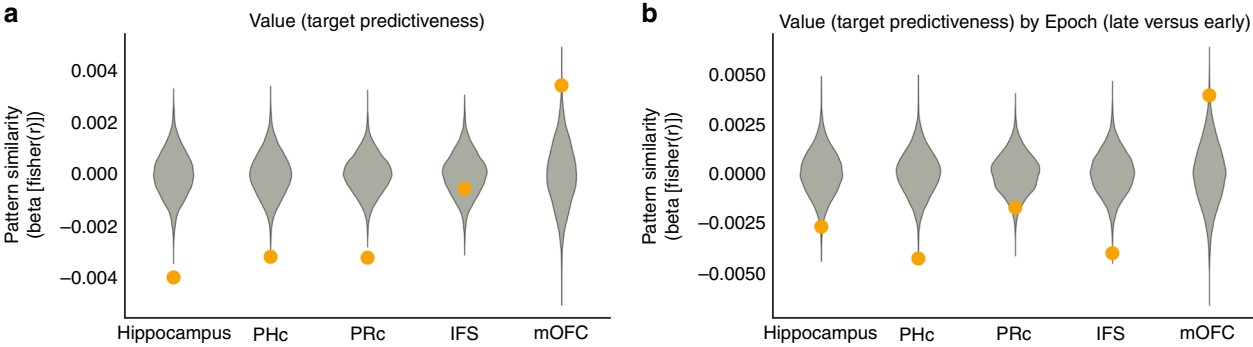

**Fig. 5** Pattern similarity analysis of stimulus value. **a** A regression analysis on PSA matrices showed that stimuli with similar associations to the target were further apart in pattern space in the hippocampus, PHc, and PRc. In contrast, they were closer together in pattern space in the mOFC. The y-axis shows the regression weights from the value similarity regressor on the PSA matrix of each ROI. **b** Hippocampus, PHc, IFS, and PRc representations for stimuli with similar associations with the target moved further apart in pattern space over the course of a run. In contrast, in mOFC, representations became more similar. The y-axis shows the regression weights from the interaction of the value similarity regressor with a regressor that encodes comparisons between trials late in learning versus comparisons between trials early in learning. Green violins show the null distributions of regression coefficients from 10,000 randomly permuted PSA matrices. Source data are provided as a Source Data file

strength of their association with the target, Fig. 5a. We used the values from the Value Spread model to compute a value regressor that was higher when stimulus-target associations were more similar. This regressor was negatively related to hippocampal pattern similarities, which shows that the hippocampus represented stimuli with similar values more distinctly, $p < .001$, FDR corrected, permutation test. We also observed this relationship in the PRc, $p < .001$ and PHc, $p = .001$, FDR corrected, permutation test. In contrast, the mOFC showed a positive effect of value, representing stimuli with similar association to the target more similarly, $p = .001$, FDR corrected, permutation test. Similar to the overlap analysis, we also found a marginal effect of higher striatal conjunctive PE on runs in which trials with similar outcomes are represented more dissimilarly in the hippocampus, $t(31) = -2.24$, $d_z = -0.40$, $p = .063$, and PRc, $t(31) = -2.48$, $p = .063$, $d_z = -0.45$, FDR corrected, permutation test.

Recent work has shown that learning drives representations of stimuli with similar outcomes apart in the hippocampus, resulting in representations of similar items that are even more distinct than representations of unrelated items[35,36]. We constructed a new model with an additional interaction term between the stimulus value and an epoch regressor, which was positive for comparisons between stimuli late in the run and negative for comparisons between stimuli early in the run. We found an interaction in the hippocampus, $p = .01$, FDR corrected, permutation test, such that the pattern distance between stimuli with similar values increased over the run (Fig. 5b). We also observed this effect in PHc, $p < .001$, IFS, $p < .001$ and PRc, $p = .035$, FDR corrected, permutation tests. In contrast, in mOFC, we found the opposite: the pattern distance for stimuli with similar values decreased over the run, $p = .01$, FDR corrected, permutation tests.

**Pattern content analysis**. The previous analyses show the hippocampus had the most distinct representations of stimuli that shared features among our regions of interest. However, the demonstration of no significant increase in similarity of hippocampal representations for feature-sharing stimuli begs for a more direct test of pattern separation. To directly test this hypothesis, we probed the content of the hippocampal and cortical ROI representations using estimates of categorical feature coding acquired from independent localizer data. If hippocampal conjunctive representations are pattern separated from their constituent features, then they are not composed of mixtures of representations of those features[37,38], Fig. 6a. Unlike high-level sensory cortex, the hippocampal representation of {face and house} would not be a mixture of the representation of {face} and {house}[37,39]. We predicted that the hippocampal representations of two-feature stimuli (e.g., {face and house} trials) in our learning task should be dissimilar from representations of faces and houses in the localizer. Hippocampal representations of one-feature trials (e.g., {face} trials), which are less conjunctive because they contain only one task-relevant feature, should be more similar to representations of the same one-feature category (e.g., faces) in the localizer. In contrast, the cortical representations of both two-feature and single-feature trials should be similar to representations of their corresponding features in the localizer, Fig. 6a. We predicted that the hippocampal representations would be less similar to feature templates than cortical ROIs, and that only the hippocampus would show would less similarity for two-feature than single-feature trials.

We correlated the patterns in each ROI with the corresponding localizer feature templates (see Methods) but were unable to detect reliable feature responses from IFS or mOFC in the localizer data[40]. We found significant similarity among task patterns and feature templates for all conditions, $p < .001$, permutation test, FDR corrected, except for hippocampal responses to conjunctive stimuli, $p = .116$, permutation test, FDR corrected, Fig. 6. As expected, the hippocampus had lower similarity to feature templates than PRc, $p < .001$, and PHc, $p < .001$, permutation test, Supplementary Note 11, 12. This effect was not likely driven by regional signal quality differences, as the hippocampus had the strongest within-stimulus coding, Fig. 4a. The hippocampal feature template was more similar to the response to a single-feature stimulus than to a two-feature stimulus, $p < .001$, permutation test, consistent with a gradient in pattern separation as the number of task-relevant features increased. This effect was larger in the hippocampus than either PRc, $p < .001$, or PHc, $p < .001$, permutation test. We confirmed these results using a parametric mixed effects analysis and also performed a control analysis to verify the results were not driven by stimulus-general activation, Supplementary Figure 6, Supplementary Note 13. Unexpectedly, we also observed that the similarity in PRc and PHc was stronger for two-feature than for single-feature stimuli, both $p < .001$, permutation test. In the mixed effects model, this result was marginally significant in PRc and nonsignificant in PHc and should be interpreted with caution.

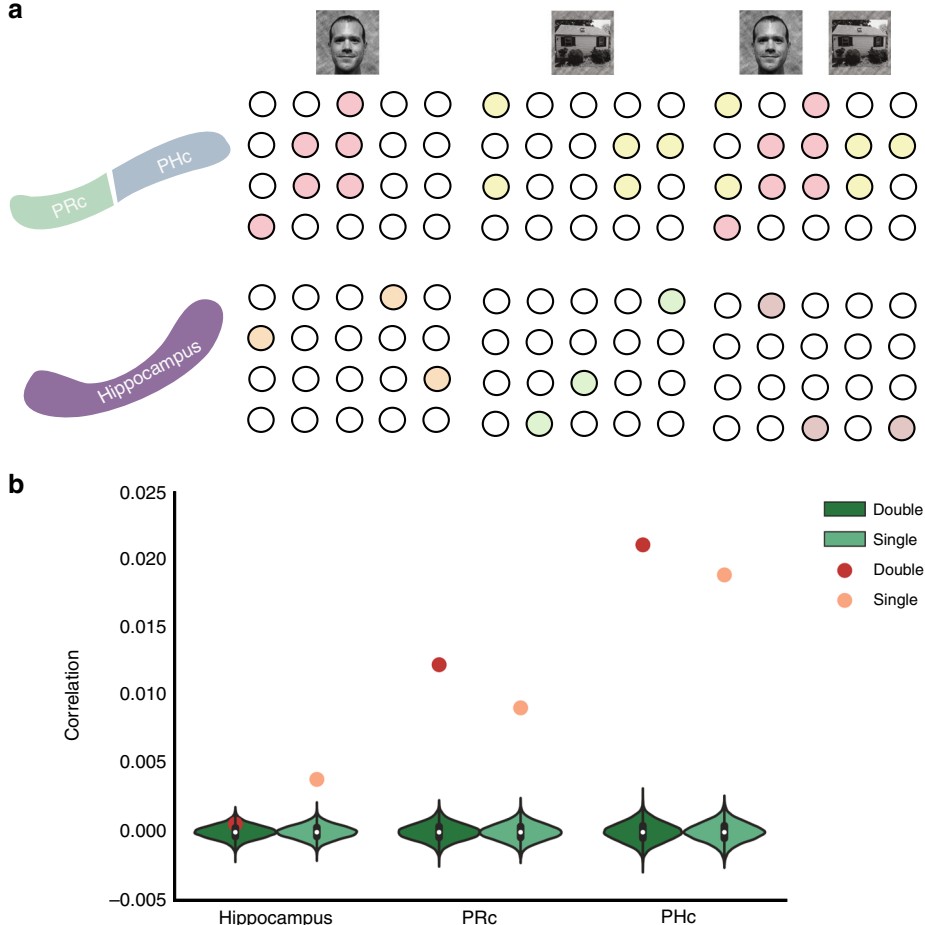

**Fig. 6** Pattern content analysis. **a** Neural predictions: top panel shows putative neural ensembles in high-level sensory cortex (parahippocampal, PHc; perirhinal, PRc) for task stimuli. Two-feature stimulus should be represented as union of responses to component features. The lower panel shows putative neural ensembles in the hippocampus; the neural representation of two-feature conjunctions should be orthogonal to responses to its component features. **b** Hippocampal patterns were less similar to feature templates than PRc and PHc representations, consistent with increased conjunctive coding. In the hippocampus, pattern similarity to templates was higher for single-feature than two-feature stimuli, consistent with increased pattern separation for stimuli with multiple task-relevant features. PRc and PHc showed increased similarity for two-feature relative to one-feature stimuli. Source data are provided as a Source Data file

## Discussion

We found evidence that pattern separation in the hippocampus enabled learning of stimulus-outcome relationships over multi-featural stimuli. Our finding of overlapping representations of conjunctions that shared features in PRc and PHc, in combination with our finding of mixed conjunctive and feature learning in both behavior and the striatal error response, suggest that feature representations influenced learning even when they did not benefit performance. The striatum receives inputs from diverse cortical and subcortical areas and might integrate predictions from systems that represent the environment in different ways (e.g., conjunctive versus feature)[41,42]. Over longer training periods it seems reasonable that subjects would learn to down-weight the influence of cortico-striatal synapses representing uninformative features. Attention could support this process by influencing the relative strength of representations of conjunctions versus features[24].

Lesion studies showing dissociations between the hippocampus and striatum in learning[43] along with some imaging studies demonstrating a negative relationship between hippocampal and striatal learning signals[44] have led to the hypotheses that these regions compete during learning and that learning transfers from hippocampal to striatal systems over time. By contrast, other evidence suggests cooperative interactions. For example, neurons in the striatum represent spatial information derived from hippocampal inputs[45] and contextual information in the hippocampus drives the formation of conditioned place preferences via its connection to ventral striatum[41,46]. These findings, in concert with our own, support a model in which hippocampal information about spatial contexts, location, or conjunctions serves as inputs for striatal associative learning (see Supplementary Discussion).

Our findings that the hippocampus represented stimuli with similar outcomes more differently and that this pattern distance increased during learning is consonant with recent demonstrations of repulsion of hippocampal representations of related memories in navigation[36] and relational learning[35,47,48]. These findings, together with our results, suggest the hippocampus dynamically increases the representational distance between overlapping experiences that could otherwise be subject to integration in cortical circuits. However, our results are inconsistent with a related study that found that the hippocampal representation of conjunctions contained information about constituent features[49]. In their task, subjects were required to integrate the value of two objects to imagine the value of their conjunction. The inconsistency could arise if task demands

determine whether the hippocampus integrates across or distinguishes between related experiences. Goal-directed attention could play an important role in determining which aspect of hippocampal function comes to dominate the hippocampal representation[50].

Our results complement and extend a recent investigation of the role of the hippocampus in conjunctive learning[51]. In this task, subjects were required to learn the relationship between conjunctions of cues and a weather outcome. They observed a univariate relationship between the hippocampal BOLD response and the degree of conjunctive learning, as well as a correlation between hippocampus and nucleus accumbens that relates to conjunctive learning. In addition to these univariate effects, they also observed a within-stimulus similarity effect in the hippocampus but did not investigate stimulus overlap nor the representations of other cortical ROIs. Our results therefore provide a unique demonstration that the hippocampal code is more pattern separated than other cortical ROIs during conjunctive learning.

We did not find evidence for the coding of stimulus identity in the mOFC; however, we found the mOFC represented stimuli with similar strength associations to the target more similarly. These findings are inconsistent with the strongest version of a recent theory suggesting that the OFC supports a state space representation in learning tasks[21]. However, it is possible that OFC represents just two states corresponding to different action policies ({AB+, C+} versus {AC−, B−}). We prefer the interpretation that the mOFC representations in our task were driven by the subjective value of the stimuli, because the target was associated with the possibility for reward[18] Finally, our distinct findings between mOFC and hippocampal representations echo recent investigations of context-based decision in rodents showing that the hippocampal representation is primarily driven by the context[52], whereas the orbitofrontal cortex representation is primarily driven by reward value[53].

The circuit properties of the hippocampus allow it to rapidly bind distributed cortical representations of features into orthogonalized conjunctive representations. Dense inputs to the striatum suggest hippocampal representations could also form the basis for associative learning over conjunctive codes. Our results extend the role of the hippocampus to include building conjunctive representations that are useful for striatal outcome and value learning.

## Methods

**Experimental model and subject details**. The study design and methods were approved by and followed the ethical procedures of the Stanford University Institutional Review Board. Forty subjects provided written informed consent. Data from eight subjects were excluded from analyses: One ended the scan early due to claustrophobia; three had scanner-related issues that prevented reconstruction or transfer of their data; two had repeated extreme (>2 mm) head movements across most runs; and two subjects demonstrated extremely poor performance, as indexed by less than $2.50 of earnings (see below for payment details). Note that our task was calibrated to the individual subjects' practice data in such a way that a simple target detection strategy would be expected to earn $7.50, and any effort to learn the task should improve on these earnings. There were 32 subjects in the analysis cohort, 19 females, mean age 22.1 years, SD 3.14, range 18–29. Due to an error, the behavioral data for one subject were lost; thus, while this imaging data were included in fMRI analyses, all behavioral analyses were conducted with a sample of 31 subjects.

**Task**. Subjects performed a target detection task in which performance could be improved by learning predictive relationships between visually presented stimuli and the target. The target appeared 70% of the time for two-feature stimulus AB as well as the single-feature stimulus C, and 30% of the time for two-feature stimulus AC as well as the single-feature stimulus B. The task bore strong similarities to the "ambiguous feature discrimination task" used to study rodent learning[23]. Subjects were instructed that they would earn 25¢ for each correct response, lose 25¢ for each incorrect response, or earn no money for responses that were slower than threshold or missed.

Response time (RT) thresholds were calibrated for each subject so that responses initiated by the perception of target onset lead to success on 50% of trials. This procedure incentivized subjects to learn predictive information to respond more quickly. Before scanning, the subjects performed a simplified target-detection trial in which they responded to a probabilistic target with no predictive relationships between the cue and target. During this session, RT thresholds were adjusted by 30 ms increments on each trial (fast-enough responses reduced the threshold while too-slow responses increased it). During the task, we continued to make smaller changes (10 ms) so that the threshold could change if subjects became progressively faster. Earning rewards on more than 50% of trials required anticipating the target onset based on the preceding stimulus (A, B, AB, or AC). The subjects were instructed that in order to earn the most money, they should learn which stimuli predicted the target and respond as quickly as possible, even if the target had not yet appeared. These instructions were meant to bias subjects towards an instrumental learning strategy, rather than an explicit rule-based learning strategy[31].

Subjects performed one practice run and were instructed on the types of relationships they might observe. During fMRI scanning, subjects performed three runs of the task. Each run consisted of 10 trials for each stimulus (AB+, AC−, B−, C+), resulting in 40 total trials per run. Features A, B, and C were mapped to a specific house, face and body part image, respectively, for the duration of the run. Subjects were not pre-exposed to the specific stimuli. The category-to-stimulus mapping was counterbalanced across runs, resulting in each visual category being associated with each feature type (A, B, or C) over the course of three runs. The counterbalancing of category-to-stimulus mapping ensured that any carry-over effects of learning across runs only had a detrimental and noisy effect on learning that worked against our hypotheses. In addition, we verbally emphasized that mappings changed between runs. Further, different participants saw different stimuli within each category and all subjects saw different stimuli across runs. Each subject encountered the same pseudo-random trial sequence of both stimuli and targets, which facilitated group modeling of parametric prediction error effects. Features appeared on either the left or the right of a fixation cross, with the assignment varying randomly on each trial. For single-feature stimuli, the contralateral location was filled with a phase-scrambled image designed to match the house/face/body part features on low-level visual properties. The target stimulus was a car image and was consistent across all trial types. On trials in which the target appeared, it did so 600 ms after the onset of the visual cues. Inter-trial intervals and the interval between the stimulus/stimulus + target and feedback were taken from a Poisson distribution with a mean of 5 s, truncated to have a minimum of 2 s and maximum of 12 s. The visual localizer task details are described in the Supplementary Methods.

**Behavioral analysis**. We used reaction time data from subjects to infer learning in the task, an approach that has been used successfully in a serial reaction time task[54]. Log-transformed reaction times were fit with linear regression, with the difference that we jointly fit the parameters of the regression model and the parameters of a reinforcement learning model. Specifically, we modeled reaction time with a value regressor taken from a reinforcement learning model:

$$V(s)^{t+1} = V(s)^t + \alpha[R_t - V(s)^t], \tag{1}$$

where $R_t$ is an indicator on whether or not the target appeared, α is the learning rate, and $s$ is the state. These values represent the strength of association between a stimulus and a target/outcome, and are not updated based on the reward feedback, which also depends on whether the subject responded quickly enough. The values we measured are more relevant for learning because they correspond to the probability that the subject should respond to the stimulus. We constructed values from three different models that made different assumptions about the underlying task representation. For the Conjunctive and Value Spread models, the state corresponded to the current stimulus $s \in \{B, C, AB, AC\}$. For the Feature model, the states were single features $s \in \{A, B, C\}$ and in two-feature trials (e.g., AB), the value was computed as the sum of the individual feature values (e.g., $V(AB) = V(A) + V(B)$), and both feature values were updated after feedback. Finally, the Base Rate learning model maintained a single state representation for all four stimuli, Supplementary Note 3.

The Value Spread RL model was a variant of the conjunctive model in which a portion of the value update spreads to the overlapping stimulus:

$$V(s')^{t+1} = V(s')^t + \alpha[R_t - V(s)^t] * \omega O(s, s'), \forall s' \neq s, \tag{2}$$

where $O(s, s')$ is an indicator function that is equal to 1 when the two stimuli share a feature and 0 otherwise, and ω is a spread parameter that controls the magnitude of the spread. For example, if the current stimulus was AB, a proportion of the value update for AB would spread to B and to AC. We allowed value to spread among any stimuli sharing features (e.g., an AB trial would lead to updates of both AC and B). This approach reflects the fact that not only will conjunctions activate feature representations in cortex, but features can activate conjunctive representations[55], a property that has been extensively studied in transitive inference tasks[55]. We fit the models using Scipy's minimize function. We calculated

likelihoods from the regression fits using the standard regression formula that assumes normally distributed errors ($\sigma^2$):

$$LL = n\log\left(\frac{1}{\sqrt{2\pi\sigma}}\right) - \frac{1}{2\sigma^2}\sum_{t=1}^{n}(rt^t - \sum_k \beta_k V_k^t)^2, \qquad (3)$$

where $n$ is the number of trials, the $V_k^t$ are $k$ different estimates of value at trial $t$, and the $\beta_k$ are regression coefficients. Finally, we fit a null No Learning model with no value regressor (and therefore fits to only mean RT). The model comparison procedures are described in detail in the Supplementary Methods.

In order to analyze the relationship between striatal BOLD and model-derived prediction error, we modeled the Feature model prediction error as well as the difference between the Conjunctive model prediction error and the Feature model prediction error. This subtraction approach has three advantages: (1) it reduces the shared variance considerably from modeling the two prediction errors (feature and conjunction, $r(118) = .79$, Note that the correlation flips sign because of the subtraction); (2) it provides a stronger test about whether conjunctive representations contribute to striatal error responses, because that contribution must be over and above the contribution of the Feature learning model; and (3) together, the two regressors combined are a first-order Taylor approximation to a hybrid model that weighs contributions of a conjunctive and a feature learning mechanism. This hybrid model is conceptually similar, although not identical to, the Value Spread model. This similarity means that we were able to model the data by supposing a hybrid mechanism, as we found in behavior, while also separately examining components of that hybrid mechanism.

In order to relate behavior to neural data, we sought to calculate an index of conjunctive learning on a per-run basis because we were underpowered to detect across-subject correlations with our sample size. We calculated the likelihood of the data for each run and subject under the maximum likelihood parameters for the Value Spread model and the Conjunctive model. We then computed the difference in likelihood between the models. This measure reflects the extent to which the Conjunctive model accounts for the data better than the Value Spread model. In order to partial out individual differences in engagement in the task, we computed the difference in likelihoods between the Value spread and the No Learning model and entered this difference as a nuisance regressor in our mixed effects models.

**fMRI modeling**. The fMRI acquisition and preprocessing as well as ROI selection procedures are described in detail in the Supplementary Methods, and ROIs are depicted in Supplementary Figure 3. The analysis was conducted using an event-related model. Separate experimental effects were modeled as finite impulse responses (convolved with a standard hemodynamic response function). We created separate GLMs for whole brain and for pattern similarity analyses (PSA). For the whole brain analysis, we modeled the (1) stimulus period as an epoch with a duration of 2 s (which encompasses the stimulus, target and response) and (2) the feedback period as a stick function. In addition, we included a parametric regressor of trial-specific prediction errors extracted from a Feature reinforcement learning model that learns about A, B, and C features independently, without any capacity for learning conjunctions. In addition, we included a regressor that was computed by taking the difference between the Feature RL errors and the Conjunctive RL model errors. This regressor captured variance that was better explained by conjunctive RL prediction errors than by feature RL prediction errors. Both parametric regressors were z-scored and were identical across all subjects. We included nuisance regressors for each slice artifact and the first principal component of the deep white matter. Following standard procedure when using ICA denoising, we did not include motion regressors as our ICA approach is designed to remove motion-related sources of noise. GLMs constructed for PSA had two important differences. First, we did not include parametric prediction error regressors. Second, we created separate stimulus regressors for each of the 40 trials. Models were fit separately for individual runs and volumes of parameter estimates were shifted into the space of the first run. Fixed effects analyses were run in the native space of each subject. We estimated a nonlinear transformation from each subject's T1 anatomical image to the MNI template using ANTs. We then concatenated the functional-to-structural and structural-to-MNI transformations to map the fixed effects parameter estimates into MNI space. Group analyses were run using FSL's FLAME tool for mixed effects.

**PSA analysis**. We were interested in distinguishing the effect of different experimental factors on the pattern similarity matrices (PSM). PSM preprocessing is described in the Supplementary Methods. We constructed linear models of each subject's PSM. We included main regressors of interest as well as several important regressors that controlled for similarities arising from task structure. The main regressors of interest were (1) a "within-stimulus similarity" regressor that was 1 for pairs of stimuli that were identical and 0 otherwise and (2) an "overlap" regressor that was coded as 1 for pairs of stimuli that shared features, −1 for those that did not, and 0 for pairs of the same stimuli. We also included exploratory regressors of interest (3) a "prediction error" regressor that was computed as the absolute value of the difference in trial-specific prediction errors, extracted from the Value Spread model, between the two stimuli, Supplementary Figure 5, 4) a

"value" regressor that was computed as the absolute value of the difference in trial-specific updated values, extracted from the Value Spread model, between the two stimuli, Fig. 5. We included two nuisance regressors that model task-related sources of similarity that are not of interest: (5) a "response" regressor that was coded as 1 for stimuli that shared a response (both target or both non-target) and −1 otherwise, (6) a "target" regressor that was coded as 1 for stimuli that both had a target, −1 for both non-target, and 0 otherwise. Finally, we included nuisance regressors for (7) the mean of the runs and (8) two "time" regressors that accounted for the linear and quadratic effect of time elapsed between the presentation of the two stimuli and (9) the interaction between the time regressor and the within-stimulus similarity regressor. We included this last interaction because the within-stimulus similarity effects were by far the most prominent feature of the PSA, and the temporal effects were therefore likely to have larger effects on this portion of the PSMs. We included prediction error (3) and value (4) regressors because we were interested in exploratory analyses of these effects based on theoretical work suggesting that the hippocampus pattern separates stimuli based on the outcomes they predict[32]. Both the value and the prediction error regressors were orthogonalized against the response and target regressors, thereby assigning shared variance to the regressors modeling outcomes. All regressors were z-scored so that their beta weights could be meaningfully compared. In addition, we estimated two additional models in which we replaced the overlap term (a difference regressor) with regressors coding for feature-sharing stimuli and non-feature-sharing stimuli separately. These regression weights were plotted in Fig. 4b for visualization purposes only.

Correlations, our dependent variable, were Fisher transformed so that they followed a normal distribution. To assess the significance of the regression weights as well as differences between regions, we compared empirical regression weights (or differences between them) to a null distribution of regression weights (or differences between them) generated by shuffling the values of the PSA matrices 10,000 times. The permutation test p-values are one-sided. All other results of parametric tests in the manuscript are two-sided. In addition, we fit a linear mixed effects model using R with subject as a random intercept, ROI as a dummy code with hippocampus as the reference, and interactions with ROI for each of the above regressors. Using random slopes resulted in convergence errors, and so we did not include them. By using both a parametric and a nonparametric approach to assessing our data, we gained confidence that our results are robust to differences in power between different statistical analysis techniques due to outliers or violations of distribution assumptions.

**Pattern content analysis**. Data from the localizer task were preprocessed and analyzed in the same manner as the main task data. GLMs were constructed for each run and included a boxcar regressor for every miniblock with 4-s width, as well as a nuisance regressor for the targets, each slice artifact and the first principal component of the deep white matter. To compute template images, we computed the mean across repetitions of each stimulus class (face, place, character, object, body part). For the pattern content analysis depicted in Fig. 6, we computed the correlations as follows: Assume that A is a face, B is a house, and C is a body part. For "single-feature" stimuli, we computed the similarity of B trials with the house template and C trials with the body part template. For "two-feature" stimuli, we computed the similarity of AB trials with the house template and AC trials with the body part template. Therefore, within each run, the task-template correlations for AB and B (and AC and C) were computed with respect to the same feature template. Computing the task-template correlations in this way means that any differences between AB and B (or AC and C) correlations reflect differences in the task representations, rather than potential differences in the localizer representations. We repeated this computation for the stimulus category mappings in each run.

**Reporting summary**. Further information on experimental design is available in the Nature Research Reporting Summary linked to this article.

## Data availability

Raw MRI and behavioral data are available at OpenNeuro with accession code ds001590. Analysis code is available at https://github.com/iancballard/Hipp_Pattern_Separation_Code. The source data underlying Figs. 2a–f, 3b, 4a–b, 5a–b, 6b and Supplementary Figs. 1a–f, 2a–d, 4, 5a–c, 6a-b and Table 1 are provided as a Source Data file. Key resources are listed in Supplementary Table 2.

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

## Acknowledgements

We thank the NSF GRFP (ICB), NSF 0801700 and Stanford Innovation Grants (ICB). We also thank Kim D'Ardenne for significant editing and Stephanie Gagnon, Karen LaRocque, Alex Gonzales, Anna Khazenzon and Yuan Chang Leong for their feedback.

## Author contributions

Conceptualization, ICB and S.M.M.; Methodology, ICB; Software, ICB; Formal Analysis, ICB; Data curation, ICB; Writing original draft, ICB; Writing, review, and editing, S.M.M. and A.D.W.; Funding acquisition, ICB and S.M.M.; Supervision, S.M.M. and A.D.W.

## Additional information

**Competing interests:** The authors declare no competing interests.

