## [Peer Review File · Nature Communications]

Reviewers' Comments:

Reviewer #1:

Remarks to the Author:

The authors present a timely and well-motivated study of the potential use of conjunctive representations in a simple response time task. Their stimuli have either a single feature or two features that predict or anti-predict the presence of a go target to follow the features. On the basis of model comparisons as fit to response times, they conclude that subjects' behavior reflects the use of a mixture of individual-feature and conjunctive representations. They suggest that these conjunctive representations must be formed by the hippocampus. fMRI pattern analysis supports the hypothesis that hippocampus, and not prc, phc, or ifs, encodes conjunctive representations of these stimuli. Further, the degree to which hippocampus codes conjunctions as separate representations, on a run-by-run basis, was negatively correlated with the striatal BOLD response to the conjunctive reward prediction error term.

The study addresses an interesting and important question, and the statistical analyses are thorough. However, I have some questions about possible conceptual gaps in the analysis logic, and would like to better understand the data and whether the models are appropriate to it.

- Additional model comparisons. I wonder in particular whether a model slightly more intelligent than the "no learning" model might be a more justifiable null hypothesis. Specifically, a model in which subjects are learning the overall rate of target presentation, unconditional on the feature stimuli. This learner would incrementally estimate the probability of the stimulus appearing, and so would show fluctuations of over- and under- estimating the base rate. In other words, a choice or perseveration kernel, of the sort that has repeatedly been shown to contribute in simple response tasks.

- This model could fit on its own, or, I think more likely, it could be an important component of the Conjunctive RL model. This version of a hybrid model would match repeated observations of two components to learning in serial response tasks, with each component operating at different timescale (Sugrue et al; Lau & Glimcher; Miller et al). This might help explain why the omega parameter is around 0.5, which, if I am understanding it correctly, "squashes" the estimates together.

- Another possibility is a Hybrid model with a separate learning rate on feature and conjunctive representations

- What are the resulting probabilities predicted by the Hybrid model, as fit to the data? I suggest looking at trial-by-trial plots for each subject. Are there "runs" that don't seem to meaningfully track the stimulus identities? If so, I wonder how well the model is capturing true stimulus-bound learning.

- This is particularly concerning because the beta parameter appears not to be different from zero, across subjects?

- Are the Hybrid values actually correlated with RTs?

- In a purely model-free analysis, do RTs track the stimulus predictiveness?

- If there are such runs in the response times of the subjects, and they are not as stimulus-driven, then I wonder if a blocking effect might have swamped the predictiveness of the individual-feature stimuli. Were subjects pre-exposed to the stimuli? Did ordering matter - e.g. if they learned AB+ first, were they less able to learn B-, or AC-?

- If the Hybrid story is real, then individual differences in omega parameter should correspond to

individual differences in PRC/PHc and HC separability, no?

- You show a negative correlation with striatal conjunctive PE, and between striatal conjunctive PE and omega... but what about the link between hc overlap and omega?
- The HC->omega link seems to be the critical one. The others can be explained by latent factors of no interest, such as disengagement with the task or responding irrespective of contingencies:
- HC overlap should go up and str conjunc PE should go down when participants are relatively less engaged with the stimuli
- And similarly for striatal conjunctive PE (down) and omega (up)
- ... but, by your hypothesized model, shouldn't HC overlap and omega should be /negatively/ correlated?
- Relatedly: Do the corresponding results hold for the striatal feature PE?

I also have some other questions that may not bear directly on the main hypothesis:

- What was the target stimulus?

Unclear from the description and the figure what the content of the stimulus was, whether it was the same for all trials/predictive stimuli, what was its timing, etc.

The timing, in particular, seems potentially impactful for the results.

- What is the correlation between feature PE and conjunction PE?
- Did you exclude the possibility that striatum is just tracking wins and losses?
- A commonly held understanding is that response selection "transfers" over the course of the task from being driven by hippocampal learning to striatal learning.
- Might a similar pattern of transfer explain the fact that the Hybrid model prevailed, across all trials?
- What is the pattern of relative fit of Conjunctive to Feature, if you bin trials into, say, quarters or thirds?

Reviewer #2:

Remarks to the Author:

Ballard and colleagues used a feature discrimination task in which human subjects had to predict a go target based on either a single or combined stimulus. Participants had to press a button within a limited time window when they detected a target. Correct target detection led to reward. Modelling showed that a mechanism in which subjects treat conjunctive stimuli separately, but also misattribute outcomes to the constituting features to some extent (an update following a conjunctive AB trial is partly misattributed to A and B stimuli) performs best. Neural analyses showed that conjunctive stimuli were less correlated to their constituting features in the hippocampus than in cortical central areas, and that the degree of this distinctiveness was related to a stratal prediction error signals.

This is well written study that addresses an original and important issue of high interest to the field. It therefore has the potential to result in a significant advancement of our knowledge. Although I am quite sympathetic to the investigated question, I do have major concerns regarding the analysis and the suitability of the paradigm to the test the raised question.

Major points:

The difference in feature-conjunction correlations between brain areas seems somewhat unspecific. Specifically, I am wondering whether the fact that HPC has higher within stimulus similarity already explains that it has less feature-conjunction correlations. It seems plausible that a brain region with high fidelity cue/item representations would generally have less cross-cue correlations, regardless of

whether they share features or not. I would therefore like to see (1) the correlations between conjunctive cues (AB-AC) in all ROIs, (2) the correlations between individual cues (B-C) and the non-shared feature/conjunctive correlations (B-AC, C-AB). I think it is important to show that the difference between HPC and the cortical control ROIs is specific to the overlapping pairs.

Couldn't be an alternative interpretation that the neural similarity is driven by similarity of target prediction rather than features, i.e. AB+ and C+ are similar because both have stronger associations to the target, and as a consequence AB-B correlations are weaker? Is there any way to disentangle these possibilities? As mentioned above, it seems important to see the full pattern of correlations in order to understand the possible interpretations and I am doubtful that the inclusion of nuisance variables in the GLM would fully address the problem.

I am a bit skeptical that the way the feature and the conjunctive PE are calculated ensures that there is no shared variance. What is the correlation between the two regressors? Wouldn't it make sense to look at the difference between reward and the value of the shown stimulus ($R-V(s)$) versus reward minus the value of the partial feature stimulus ($R-V(s')$ if s' is part of s)?

I am concerned that the task did not require or reward progressively faster go responses with learning in the same way a standard RL task would reward more choices of a high versus low rewarding stimulus. The reason is that subjects needed to be faster than the threshold, but not generally as fast as they can (there was no benefit for faster than threshold choices). At the same time, too fast choices were dangerous as they could lead to losses. So it seems unlikely that participants would just globally minimise RT for higher rewarding cues, but this seems to be the assumption of the models.

On a similar note, it seems remarkable that the behavioural evidence in favour of conjunctive learning is mixed, while all the neural analyses are based on the assumption that subjects care about/try to learn about the conjunctive stimuli. How do the authors reconcile the neural and behavioural finding?

Relatedly, it would be important to understand participants' behaviour better. What was the average percent correct and RT for the different cues? What was the amount of false alarms versus misses? And how did the RT threshold change over time?

The relation between the PE and the pattern separation measure seems broad as it is not specific to any ROI and could reflect more general signal quality aspects. How do the other correlations (cue with same cue, feature with non-overlapping conjunctive stimulus) relate to the PE? Does the relative difference between overlapping pair correlation (AB-B) and unspecific correlations relate to PE?

A number of relevant studies have investigated the effects of overlap on hippocampal representations, for instance Chanalles et al., 2017, *Curr Biol.* or Favlia et al., 2017, *Nat Communications*. In addition, several studies have investigated the effects of overlap of features during value learning and prediction, and their effects on hippocampal representations, Barron et al., 2013, *Nature Neurosci*. It seems remarkable that Barron et al. found that HPC representations during value prediction of compound stimuli are more similar to their components than to other stimuli (i.e., the reverse of what was found here). I think these studies and their relation to the presented work should be considered. Note that I am not an author on any of the mentioned studies nor do I have close ties with the authors.

Based on the Barron study and the large literature on value and state information in ventromedial prefrontal areas, it seems interesting to investigate value signals and effects of stimulus overlap in these ROIs too, especially given the caveat noted above that the current study significantly deviates from classical value based choice tasks in meaningful ways.

I am not sure how the RTs were converted into likelihoods. Which distribution was assumed, and where parameters fixed? More details on the modelling would be helpful.

Unlike the Niv et paper, the feature RL model assumed equal weighting between features. Have the authors tried to fit feature weights too? I am almost surprised the feature RL model performs second best, given that one cannot perform well in the task when feature values are learned individually.

It would be helpful to briefly mention the number and nature of parameters for the different models during their description in the main text.

What is dz? (p11 top)

I don't fully follow the authors logic in the discussion when they state that "our data suggest future models that exploit the computational ...". Is this about AI models?

The GLMs set up to get pattern estimates that are used in the PSM seem peculiar. Why are nuisance variables for overlap and same/different target included in the analyses, when the effect overlap is part of the question? Does that come change and influence how much pattern overlap you will find? In addition, it seems unusual to have PE and value regressors, when these usually are parametric modulator assigned to a stimulus regressor.

We want to thank the editor and the reviewers for their thoughtful and helpful comments on our manuscript. We were pleased that the reviewers expressed enthusiasm about the manuscript indicating that the manuscript is a “well-motivated”, the statistical analyses are “thorough” (R1), and that it addresses an “original and important issue of high interest to the field” (R2). Both reviewers raised concerns and clarifications of our fMRI analysis and suggested control analyses, as well as helpful control analyses for behavior. We have made conducted all requested control analyses and made several major changes to the manuscript in order to address all of the reviewers’ points. We believe these changes have greatly improved the manuscript. These changes are described below.

Reviewer 1

The authors present a timely and well-motivated study of the potential use of conjunctive representations in a simple response time task. Their stimuli have either a single feature or two features that predict or anti-predict the presence of a go target to follow the features. On the basis of model comparisons as fit to response times, they conclude that subjects' behavior reflects the use of a mixture of individual-feature and conjunctive representations. They suggest that these conjunctive representations must be formed by the hippocampus. fMRI pattern analysis supports the hypothesis that hippocampus, and not prc, phc, or ifs, encodes conjunctive representations of these stimuli. Further, the degree to which hippocampus codes conjunctions as separate representations, on a run-by-run basis, was negatively correlated with the striatal BOLD response to the conjunctive reward prediction error term.

The study addresses an interesting and important question, and the statistical analyses are thorough. However, I have some questions about possible conceptual gaps in the analysis logic, and would like to better understand the data and whether the models are appropriate to it.

Comment: Additional model comparisons. I wonder in particular whether a model slightly more intelligent than the "no learning" model might be a more justifiable null hypothesis. Specifically, a model in which subjects are learning the overall rate of target presentation, unconditional on the feature stimuli. This learner would incrementally estimate the probability of the stimulus appearing, and so would show fluctuations of over- and under-estimating the base rate. In other words, a choice or perseveration kernel, of the sort that has repeatedly been shown to contribute in simple response tasks. This model could fit on its own, or, I think more likely, it could be an important component of the Conjunctive RL model. This version of a hybrid model would match repeated observations of two components to learning in serial response tasks, with each component operating at different timescale (Sugre et al; Lau & Glimcher; Miller et al). This might help explain why the omega parameter is around 0.5, which, if I am understanding it correctly, "squashes" the estimates together.

Response: We agree that it is important to examine whether response perseveration could contribute to behavior in our task. Following the reviewer’s advice, we included a base-rate estimating model as a component of our conjunctive RL model. The base-rate model learns the probability of the target using reinforcement learning -- following a target (or non-target) it updates the value of all four stimuli equally with its own learning rate parameter. We included the value estimates of this model as an additional regressor on reaction time in the Value Spread model (note that in the original manuscript, we were not consistent in labeling this model Value Spread or Hybrid. We have corrected this and adopted “Value Spread”). This addition did not significantly improve the likelihood of the Value Spread model (Wilcoxon $T = 172$, $p = .14$). However, the model did appear to give reasonable results. The regression beta on both the Value Spread RL value estimate and the Base Rate values were negative and significant (Value Spread RL $T = 84$, $p = .002$; Base Rate RL $T = 136$, $p = .028$). The omega parameter did decrease somewhat, but it stayed substantial (.41), which indicates a relatively strong mixing of Conjunctive and Feature learning, even when accounting for response perseveration effects. We conclude that a response kernel explains features of behavior in our task, but its inclusion in our model added complexity without significantly improving the quality of model fits. We suspect that more data would be necessary to separately establish these two contributions to responding. In addition, the robustness of our results to this change in modeling indicate that our model fits were not driven by unmodelled effects of response kernels. We have included these results in the manuscript.

Comment: Another possibility is a Hybrid model with a separate learning rate on feature and conjunctive representations

Response: Our Value Spread model is qualitatively very similar to such a model. The spread parameter, omega, affects how much is incidentally learned about features and can vary independently of alpha, which determines learning about conjunctions. However, this model is indeed formally distinct from a two-systems model in which a feature-RL system and a conjunctive-RL system track values (potentially with different learning rates) and then combine their predictions in a weighted fashion (similar to a model-based/model-free architecture). This model and our Value Spread model differ in whether value mixing happens at the time of the value update or at the time of prediction.

We implemented the suggested mixture-model with separate learning rates for feature and conjunction representations. However, it proved very difficult to fit this mixture-model to our data. The fitting frequently failed to converge and when it did converge often gave pathological parameter estimates at the edge of the parameter space (learning rates equal to zero or one).

For the purpose of our paper's thesis, the model suggested by the review and the one we report conceptually test for the same thing: whether there is a mixture of conjunctive and feature learning in behavior. Although comparisons between the models could theoretically illuminate *how* that mixture occurs (spreading of value updates versus weighted predictions between two systems), we are not able to adjudicate between these possibilities with our data. We have added a statement to this effect in the manuscript.

Comment:

- 1) What are the resulting probabilities predicted by the Hybrid model, as fit to the data? I suggest looking at trial-by-trial plots for each subject. Are there "runs" that don't seem to meaningfully track the stimulus identities? If so, I wonder how well the model is capturing true stimulus-bound learning.
- 2) This is particularly concerning because the beta parameter appears not to be different from zero, across subjects?
- 3) Are the Hybrid values actually correlated with RTs?
- 4) In a purely model-free analysis, do RTs track the stimulus predictiveness?
- 5) If there are such runs in the response times of the subjects, and they are not as stimulus-driven, then I wonder if a blocking effect might have swamped the predictiveness of the individual-feature stimuli. Were subjects pre-exposed to the stimuli? Did ordering matter - e.g. if they learned AB+ first, were they less able to learn B-, or AC-?

Response: We thank the reviewer for raising these important points. In response to the reviewers specific questions:

- 1) See 5, below
- 2) The beta term is indeed significantly negative, indicating that faster reaction times are associated with higher values (Wilcoxon $T = 64$, $p < .001$). This result was originally described in the Methods; we have moved it to the Results.
- 3) The average correlation between the best-fit values from the Value Spread Model and reaction times was .43, which was significantly different from zero, $t(30) = 20$, $p < .001$, Fisher corrected for normality. We have added this to the manuscript.
- 4) This is an excellent question / suggestion. We have included a detailed model free analysis in the manuscript, which we briefly summarize after (5).
- 5) The comment raises concerns about possible carry-over and blocking effects. We should clarify relevant features of our design that were not sufficiently emphasized in the original submission: a) Subjects were not pre-exposed to any of the stimuli; b) a single visual exemplar, such as a specific face, was used for each stimulus and this exact stimulus was used over the course of a run; c) these exemplars were drawn randomly without replacement between runs, so, for example, subjects never saw the same face between runs; and d) the visual category to stimulus mapping was balanced across runs, so "A" was a face, body part, and house once across

the three runs. All of these manipulations, in addition to a clear demarcation between runs, were intended to eliminate or at least minimize blocking and other carry-over effects between runs. In particular, the remapping in (d) ensures that any potential value assigned to the stimulus category would not have a systematic effect across runs. That is, if a face is C in run 1, a different face may be A or B on the next run. Because of this, carry-over cannot have any beneficial learning effect, either for a feature or a conjunctive system, since any carried over value would be uncorrelated with the to-be-learned value. This would serve to add noise to the response times that is unfit by any of our models (i.e. by adding variance to initial values). This noise is equivalent across stimulus types and therefore can only work against our effects.

The reviewer proposes that we examine our data for effects of blocking or carry-over by looking at trial-by-trial plots by run and subject. Unlike in a choice task, we do not compute a likelihood for each individual trial. Rather, the likelihood is computed from a linear regression on reaction time across all trials. We took the best fit parameters for each subject (fit to all runs), and examined the likelihood of the data separately for each run with those best-fit parameters. The figure below shows this analysis, with individual lines corresponding to subjects. The likelihood has been normalized by the number of responses for each run and subject. Whereas blocking or carry-over effects would reduce our model's performance over runs, the data show a small, but significant improvement in model-fit across runs, $Z = 2.12$, $p = .034$, mixed effects model.

Finally, we followed the reviewer's excellent suggestion and conducted a model-free analysis of behavior. Consistent with conjunctive learning, subjects were faster in responding to stimuli associated with the target (AB+ and C+) than those that were not (AC- and B-), $F(1,26) = 13$, $p = .001$, $\eta_G^2 = .08$, Panel B. As a result of faster RTs, subjects had a higher hit rate for target stimuli, $F(1,26) = 43$, $p < .001$, $\eta_G^2 = .35$, Panel A. In addition, subjects were more likely to make false

alarms to target predictive stimuli when the target did not appear, $F(1,26) = 34, p = .001, \eta_G^2 = .14$, Panel C.

Our behavioral analysis showed a main effect of target, which is consistent with Conjunctive learning and cannot be explained by Feature learning, Panel D. However, we observed an additional main effect of the number of features (single versus double). Subjects were faster, $F(1,26) = 7.6, p = .01, \eta_G^2 = .02$, had a higher hit rate, $F(1,26) = 27, p < .001, \eta_G^2 = .11$, and made more false alarms, $F(1,26) = 7.9, p = .008, \eta_G^2 = .01$, for two feature stimuli. Conjunctive learning cannot account for this effect, whereas Feature learning predicts this main effect, Panel D. This is because our models were initialized with zero values, which introduces an initial bias towards learning from target appearance relative to target non-appearance that disappears over time. This initialization improved the fit of all models. Because of this, the A feature in the Feature model has positive value despite being non-predictive of reward, which causes a higher total value for conjunctions. Therefore, behavioral performance shows signatures of both Conjunctive and Feature learning.

Finally, we also observed an interaction between target and the number of features, such that subjects showed an especially higher hit rate, $F(1,26) = 9.7, p = .004, \eta_G^2 = .05$, and higher false alarm rate, $F(1,26) = 15, p < .001, \eta_G^2 = .03$, for AC- relative to B- trials. Only the Value Spread model, which mixes Feature and Conjunctive learning, can account for this interaction, Panel D. This interaction occurs in the Value Spread model because the bias towards learning about targets rapidly overwhelms the difference in AB+ and C+ due to feature learning, whereas the relatively slower initial learning about non-targets emphasizes the difference between AC- and B-. In sum, the Value Spread model accounts for the qualitative features of the data better than the Conjunctive or Feature model. Note that the models have *not* been modified from the original submission to capture these qualitative effects. This analysis has been added to the manuscript.

Comment: If the Hybrid story is real, then individual differences in omega parameter should correspond to individual differences in PRc/PHc and HC separability, no?

- You show a negative correlation with striatal conjunctive PE, and between striatal conjunctive PE and omega... but what about the link between hc overlap and omega?

- The HC->omega link seems to be the critical one. The others can be explained by latent factors of no interest, such as disengagement with the task or responding irrespective of contingencies:

- HC overlap should go up and str conjunc PE should go down when participants are relatively less engaged with the stimuli

- And similarly for striatal conjunctive PE (down) and omega (up)

- ... but, by your hypothesized model, shouldn't HC overlap and omega should be /negatively/ correlated?

Response: We agree with the reviewer's comment. Unfortunately, we are not well-powered to detect across-subjects (i.e., individual difference) correlations with our cohort size (e.g., we have only 37% power to detect a medium-sized correlation of .3). For this reason, we relegated reporting of across-subject correlations to the Supplement, as we cannot have high confidence in them with this sample size.

In terms of the suggested *individual difference* test, we would predict a *positive* correlation between hippocampal overlap and omega (because more omega means more value mixing). However, we did not observe a significant correlation between omega and the hippocampal overlap term. We describe this and other null effects in the Null Effects section of the Supplement.

We agree that the HPC overlap term and the striatal conjunctive term may both vary with attention, that this is a concern, and that it merits investigation. We ran a control analysis relating hippocampal overlap to striatal conjunctive PE. We added a nuisance variable meant to capture fluctuations in attention as fluctuations in behavioral learning performance. This variable was constructed as the difference between the likelihood of the Value Spread model and the Null model on each run. We found that the relationship between hippocampal overlap and striatal conjunctive PE persisted, $Z = -2.04$, $p = .045$. Therefore, this relationship is robust to individual differences in how well their behavior is explained by our learning model. This finding suggests that the results are not wholly due to differences in participant engagement. However, it does not exclude engagement as a possible contributor to individual differences and we have included consideration of this caveat in the Discussion.

In order to further assess brain-behavior relationships, we conducted two new analyses. We computed an index that measures the extent to which conjunctive learning was used in each run of the subject's behavior. Specifically, we computed the likelihood of the data for each run and

subject under the maximum likelihood parameters for the Value Spread model and the Conjunctive model. We computed the difference in likelihood between the models. This measure reflects the extent to which the Conjunctive model accounts for the data better than the Value Spread model. We fit a mixed-effects model of this measure with random intercepts for subjects and included a nuisance covariate that measured how well subjects learned in each run, relative to chance. This helps to ensure that the relationship is not due to fluctuations in participant engagement. Contrary to our predictions, we did not find any relationship between overlap in any of our ROIs and this measure. It is possible that variability between runs in this measure may be too large to detect fine scale relationships with behavior. We next examined whether conjunctive learning was related to univariate signal magnitude, and found that runs with stronger hippocampal activity were also runs with the most conjunctive learning, $t(31) = 3.1$, $p = .01$, $d_z = 0.56$, FDR corrected. This relationship was nonsignificant in the other ROIs, all $p > 1$. Therefore, univariate signal in the hippocampus is predictive of conjunctive learning. Relating to the reviewer's concern about attention: This finding does not rule out a model in which attention causes both hippocampal signal to increase and conjunctive learning to increase. However, the nonsignificant finding in the other ROIs suggests that the result is not merely driven wholesale differences in engagement with the task. Rather, attention could be a latent mechanism that increases hippocampal engagement with the task in order to improve conjunctive learning (Aly and Turk-Browne 2016). Future work is necessary to establish the role of attention. We have added discussion of this result and its relationship to attention to the manuscript.

Comment: - Relatedly: Do the corresponding results hold for the striatal feature PE?

Response: We note that the conjunctive PE represents variance explained over-and-above the effect of feature PE and is therefore a more sensitive measure of the degree of conjunctive PE coding in the hippocampus. As expected, there was no relationship between hippocampal overlap and the striatal feature PE, $p > .2$, and this finding suggests that general signal quality fluctuations did not contribute to the effect. We have added this result to the manuscript.

Comment: I also have some other questions that may not bear directly on the main hypothesis:

- What was the target stimulus?

Unclear from the description and the figure what the content of the stimulus was, whether it was the same for all trials/predictive stimuli, what was its timing, etc.

The timing, in particular, seems potentially impactful for the results.

Response: We apologize for this omission and thank the reviewer for catching it. The target stimulus was a car and was consistent across all trial types. On trials where the target appeared, it did so 600ms after the onset of the visual cues. We have added this information to the manuscript.

Note that the car is completely arbitrary. We explained the task to participants with a story analog in which they were searching for a car. However, the nature of the picture itself was not important and the story analogy did nothing more than to reinforce that stimuli differentially predict the subsequent appearance of a target.

Comment: What is the correlation between feature PE and conjunction PE?

Response: The feature and conjunction PEs were strongly correlated ($r(118) = .79$). However, our model used the feature PE as a regressor and the difference between the conjunction PE and feature PE as the second regressor (see discussion about our motivation for this choice on page 14, below). This procedure reduced, but did not eliminate, the magnitude of the correlation ($r(118) = -.59$). Note that the correlation changes signs because of the subtraction. Although this correlation is still large, shared variance between regressors should only work against our ability to detect significant effects. Nonetheless, we have included a description of this in a caveat in the manuscript.

Comment: Did you exclude the possibility that striatum is just tracking wins and losses?

Response: This was not explicitly included in our model, and we now address this important concern as follows. Because we constructed the conjunctive PE regressor as a difference, this assigns much of the variance attributable to outcome to the feature PE regressor. In addition, this subtraction induces an anticorrelation between the conjunctive regressor and the win/loss outcomes ($r = -.28$). Because of this anticorrelation, a *positive* striatal response to the conjunctive regressor cannot be explained by a *stronger* striatal response to wins versus losses. In contrast, the feature PE response could be partially attributable to outcome coding. To address this possibility, we extracted single trial betas from our striatal ROI (the leave-one-subject out feature PE ROI), and confirmed using a mixed-effects model with random-intercepts for subjects that both the reward outcome ($Z = 56, p < .001$) and the feature PE ($Z = 4.7, p < .001$) contribute to the striatal outcome response. This control analysis has been added to the manuscript.

Comment: A commonly held understanding is that response selection "transfers" over the course of the task from being driven by hippocampal learning to striatal learning.

- Might a similar pattern of transfer explain the fact that the Hybrid model prevailed, across all trials?

- What is the pattern of relative fit of Conjunctive to Feature, if you bin trials into, say, quarters or thirds?

Response: We thank the reviewer for the suggested analysis. Given the small number of trials per block (20 with responses), we binned runs into early and late epochs, collapsing across runs. We examined the likelihood of the data for each epoch under the MLE parameter estimates for

each model. We found an overall effect of epoch, $Z = 4.7$, $p < .001$, such that behavior was better fit by the models for later trials. However, we found no evidence of an interaction of model type by epoch, $p > .2$. These results indicate that performance of the Value Spread model likely does not reflect a transition from Feature to Conjunctive learning. In addition, our neural data support a model in which the striatum and hippocampus cooperate to support learning. We have included these results and extended our discussion of the transfer model in the manuscript.

Our work adds to an already ambiguous literature on the transfer model. Early work showed deficits in probabilistic selection only *late* in learning (which would appear to suggest a transfer in the opposite direction) (Knowlton et al. 1994). Subsequent work that established a dissociation between hippocampal and striatal learning used a shorter task design in order to isolate the period where hippocampal amnesiacs were unimpaired (Knowlton et al. 1996). More recent work has found an enduring hippocampal deficit in probabilistic learning specifically when the task involves multiple cues or task demands change (Hopkins et al. 2004; Shohamy et al. 2009). Future work is necessary to clarify the nature of hippocampal contributions to learning over different time courses and under different task demands.

Reviewer 2

Ballard and colleagues used a feature discrimination task in which human subjects had to predict a go target based on either a single or combined stimulus. Participants had to press a button within a limited time window when they detected a target. Correct target detection led to reward. Modelling showed that a mechanism in which subjects treat conjunctive stimuli separately, but also misattribute outcomes to the constituting features to some extent (an update following a conjunctive AB trial is partly misattributed to A and B stimuli) performs best. Neural analyses showed that conjunctive stimuli were less correlated to their constituting features in the hippocampus than in cortical central areas, and that the degree of this distinctiveness was related to a stratal prediction error signals.

This is well written study that addresses an original and important issue of high interest to the field. It therefore has the potential to result in a significant advancement of our knowledge. Although I am quite sympathetic to the investigated question, I do have major concerns regarding the analysis and the suitability of the paradigm to the test the raised question.

Comment: The difference in feature-conjunction correlations between brain areas seems somewhat unspecific. Specifically, I am wondering whether the fact that HPC has higher within stimulus similarity already explains that it has less feature-conjunction correlations. It seems plausible that a brain region with high fidelity cue/item representations would generally have less cross-cue correlations, regardless of whether they share features or not. I would therefore like to see (1) the correlations between conjunctive cues (AB-AC) in all ROIs, (2) the correlations between individual cues (B-C) and the non-shared feature/conjunctive correlations (B-AC, C-AB). I think it is important to show that the difference between HPC and the cortical control ROIs is specific to the overlapping pairs.

Response: The reviewer is concerned about the potential relationship between the overlap and the identity analyses. Conceptually, we agree with the reviewer, and this was in fact the purpose of doing both analyses: A region cannot have high fidelity cue representations if they are not pattern separated. However, the within-stimulus similarity [(AB, AB), (AC, AC), (B, B), (C,C)] regressor is orthogonal to the overlap regressor [(AB, AC), (AB, B), (AC, C)] versus [(AB, C), (AC, B), (B, C)], so there is no mathematical reason for the results to be interdependent. Thus, while the two analyses are conceptually related, they provide distinct empirical tests.

We agree with the reviewer that the breakdown proposed would provide a useful overall picture of the data, and we now include it below and in the Supplement. The data depict the cross-cue correlations after Fisher transform. Note that the absolute magnitude of the correlations is not meaningful because they were computed within-run (the relative magnitudes are meaningful). While illustrative, it is difficult to draw strong conclusions from visual examination from these

plots because they include a multitude of sources of similarity (response similarity, overlap, etc) due to the nature of our task. This is different from a typical hippocampal pattern similarity analysis in which the task demands are constant across all stimuli. Our regression approach separates these contributions. In addition, correlation differences in pattern similarity analyses in the hippocampus are often very small, but reliable. We strove to ensure the reliability of our results by using two different sets of statistical tests.

We interpret the reviewer comment about overlapping pairs to be about the (AB, AC) comparison. Our understanding is that the reviewer posits that the difference in HPC should be specific to the (AB-AC) comparison, and not also the (B, AB) and (C, AC) pairs. We disagree, because (AB, AC), (B, AB) and (C, AC) all share a single feature in common. Our analysis tests for difference between the hippocampus and the other ROIs is in its relative correlations for [(AB, AC), (AB, B), (AC, C)] versus [(AB, C), (AC, B), (B, C)] — we believe it is precisely this comparison that speaks to whether “the difference between HPC and the cortical control ROIs is specific to the overlapping [events]”. This analysis is specific to each region’s relative difference in pattern similarity for pairs of stimuli that share an overlapping feature.

Importantly, theories of hippocampal function propose that the hippocampus treats *all* stimuli as conjunctions, including simple stimuli composed of single features (approximated by singletons in our study). In the hippocampus, stimuli are bound to their context, including task demands, visual display properties, time, etc. This property is what endows the hippocampus with the ability to support incidental source memory. It would be more accurate to write our conditions as (ABX, ACX, BX, CX), where X refers to the (continuously changing) surrounding context. Because the A, B, and C features are the salient features in our task, AB and AC representations may be *more* similar in terms of neural representation than B and C. Therefore, rather than testing for differences between conjunctive and non-conjunctive stimuli, our overlap regressor tests for similarity in representations between conjunctive stimuli that share a salient feature versus those that do not. We have added text discussing these points in the manuscript.

Comment: Couldn't be an alternative interpretation that the neural similarity is driven by similarity of target prediction rather than features, i.e. AB+ and C+ are similar because both have stronger associations to the target, and as a consequence AB-B correlations are weaker? Is there any way to disentangle these possibilities? As mentioned above, it seems important to see the full pattern of correlations in order to understand the possible interpretations and I am doubtful that the inclusion of nuisance variables in the GLM would fully address the problem.

Response: We appreciate this concern. As detailed in the comment about the GLM below, we include nuisance variables that code for the similarity in outcomes, and some of our exploratory variables (PE and value) additionally serve to remove variance attributable to the outcome. The overlap items are (AB+, B-), (AB+, AC-) and (AC-, C+) and the nonoverlap items are (AB+, C+), (AC-, B-), and (C+, B-), so there are more pairs with the same outcome in the nonoverlap condition. Thus, there is a correlation between the overlap regressor and the effect of response ($r = .09$). This correlation is smaller than one would expect at first glance because of the stochasticity in the target occurrence. We contend that the inclusion of nuisance regressors is a standard and mathematically sound way to account for this problem. Most importantly, this correlation can only introduce a spurious increase in our overlap measure if cortical ROIs are *more* similar when the outcomes are *different* (because there are more pairs with different outcomes in the non-overlap condition). However, the observed effect of response and target is positive in all regions (Figure S5), indicating that the ROIs represent stimuli *more* similarly when the outcomes are the *same*; this effect works at odds with our main finding, and to the extent that nuisance modeling failed to fully account for it, then it would lead to an underestimate of the main finding of interest. We have included discussion of this in the main manuscript.

Comment: I am a bit skeptical that the way the feature and the conjunctive PE are calculated ensures that there is no shared variance. What is the correlation between the two regressors? Wouldn't it make sense to look at the difference between reward and the value of the shown stimulus ($R - V(s)$) versus reward minus the value of the partial feature stimulus ($R - V(s')$ if s' is part of s)?

Response: We modeled the prediction errors taken from the feature learning model as well as the difference between prediction errors taken from a conjunctive model and the feature model. The correlation is ($r(118) = -.59$). This subtraction approach was borrowed from Daw (2011), and it has three advantages: 1) it reduces the shared variance considerably from simply looking at the two prediction errors (feature and conjunction, $r(118) = .79$, Note that the correlation flips sign because of the subtraction); 2) it provides a stronger test about whether conjunctive representations contribute to striatal error responses, because that contribution must be over and above the contribution of a feature learning model; and 3) together, the two regressors combined are a first-order Taylor approximation to a hybrid model that weighs contributions of a

conjunctive and a feature learning mechanism. This hybrid model is conceptually very similar to the Value Spread model, although distinct in minor aspects.

This means that we can model the data by supposing a hybrid mechanism, as we found in behavior, while also separately examining components of that hybrid mechanism. We understand the reviewer's proposal to be to compare the prediction error of the conjunctive model (which has a $V(s)$ for each of the four stimuli) to the conjunctive error of the feature model (which computes the value of a stimulus from the sum of values of its constituent features $V(s')$). As mentioned above, this approach has a higher correlation between regressors, and does not confer the benefits described in 2 and 3.

Although the correlation between the PEs is still large (-.59), shared variance between regressors should only work against our ability to detect significant effects. Nonetheless, we now include a discussion of this caveat in the manuscript.

Comment: I am concerned that the task did not require or reward progressively faster go responses with learning in the same way a standard RL task would reward more choices of a high versus low rewarding stimulus. The reason is that subjects needed to be faster than the threshold, but not generally as fast as they can (there was no benefit for faster than threshold choices). At the same time, too fast choices were dangerous as they could lead to losses. So it seems unlikely that participants would just globally minimise RT for higher rewarding cues, but this seems to be the assumption of the models.

Response: We shared this concern and we undertook two procedures to minimize it by individually adjusting the threshold for fast enough responses:

- 1) Before scanning, subjects did a simplified target-detection trial in which they had to respond to a probabilistic target with no predictive relationships between the cue and target. During this session, RT thresholds were adjusted by 30 ms increments on each trial (fast-enough responses reduced the threshold while too-slow responses increased it).
- 2) During the main task, we continued to make smaller changes (10 ms) so that the threshold could change if subjects became progressively faster. We neglected to mention this latter component, and we thank the reviewer for raising this point so we could correct the omission. Because of both of these calibrations, and the resulting stringency of the RT threshold, subjects were incentivized to respond as quickly as they possible could in order to earn the most reward.

Below is a plot of the RT threshold over the course of a run. Dots are individual subjects and data have been average across runs. (Note that this addresses the reviewer's question about "how does the reaction time threshold change over time").

The threshold therefore decreased, which reflects participants decreasing reaction time for AB+ and C+ trials over each run, $Z = -2.04$, $p = .041$, mixed effect model with random intercepts for subjects. We now include this figure in the Supplement.

Because subjects increased the speed of their responses to target-predictive stimuli over learning, and were faster for target-predictive than target non-predictive stimuli (see new behavioral analyses below), we conclude that these task manipulations were successful in incentivizing subjects to learn predictive relationships in order to respond more quickly.

Comment: On a similar note, it seems remarkable that the behavioural evidence in favour of conjunctive learning is mixed, while all the neural analyses are based on the assumption that subjects care about/try to learn about the conjunctive stimuli. How do the authors reconcile the neural and behavioural finding?

Response: We thank the reviewer for bringing this to our attention, as our aims are to characterize how the neural data reflect the distinction we observed in behavior. We have attempted to clarify our interpretation of the results in the manuscript.

The striatal analysis shows that both kinds of learning (feature and conjunctive) contribute to the striatal PE response. Our hypothesis is that these components reflect different striatal inputs from sensory cortex and hippocampus: the sensory cortex inputs to the striatum reflect a feature learning system, whereas the hippocampal inputs represent a conjunctive learning system (with the addendum noted above that both single-feature and double-feature stimuli are represented in the hippocampus). The strength of these striatal inputs is adjusted based on feedback. Given enough trials, we expect that the contribution of feature learning would disappear. Our finding that the sensory cortical representations are more strongly feature based and the hippocampal representations are more conjunctive provide support for this model. However, we acknowledge

that our data cannot fully validate this model, and thus we limited the scope of our interpretation in the initial manuscript. To further address this point, we have extended our discussion in the manuscript so as to further clarify the relationship between the neural measures and behavior.

Comment: Relatedly, it would be important to understand participants' behaviour better. What was the average percent correct and RT for the different cues? What was the amount of false alarms versus misses?

Response: We thank the reviewer for highlighting that we can more thoroughly describe the behavior. We now include a more in-depth analysis of behavior in the main text. The updated behavioral results are summarized in this figure:

Consistent with conjunctive learning, subjects were faster in responding to stimuli associated with the target (AB+ and C+) than those that were not (AC- and B-), $F(1,26) = 13, p = .001, \eta_G^2 = .08$, Panel B. As a result of faster RTs, subjects had a higher hit rate for target stimuli, $F(1,26) = 43, p < .001, \eta_G^2 = .35$, Panel A. In addition, subjects were more likely to make false alarms following target-predictive stimuli when the target did not appear, $F(1,26) = 34, p = .001, \eta_G^2 = .14$, Panel C.

Our behavioral analysis showed a main effect of target, which is consistent with Conjunctive learning and cannot be explained by Feature learning, Panel D. However, we observed an additional main effect of the number of features (single versus double). Subjects were faster, $F(1,26) = 7.6, p = .01, \eta_G^2 = .02$, had a higher hit rate, $F(1,26) = 27, p < .001, \eta_G^2 = .11$, and made more false alarms, $F(1,26) = 7.9, p = .008, \eta_G^2 = .01$, for two feature stimuli. Conjunctive learning cannot account for this effect, whereas Feature learning predicts this main effect, Panel

D. This is because our models were initialized with zero values, which introduces an initial bias towards learning from target appearance relative to target non-appearance that disappears over time. This initialization improved the fit of all models. Because of this, the A feature in the Feature model has positive value despite being non-predictive of reward, which causes a higher total value for conjunctions. Therefore, behavioral performance shows signatures of both Conjunctive and Feature learning.

Finally, we also observed an interaction between target and the number of features, such that subjects showed a significantly higher hit rate, $F(1,26) = 9.7, p = .004, \eta_G^2 = .05$, and higher false alarm rate, $F(1,26) = 15, p < .001, \eta_G^2 = .03$, for AC- relative to B- trials. Only the Value Spread model, which mixes Feature and Conjunctive learning, can account for this interaction, Panel D. This interaction occurs in the Value Spread model because the bias towards learning about targets rapidly overwhelms the difference in AB+ and C+ due to feature learning, whereas the relatively slower initial learning about non-targets emphasizes the difference between AC- and B-. In sum, the Value Spread model accounts for the qualitative features of the data better than the Conjunctive or Feature model. Note that the models have *not* been modified from the original submission to capture these qualitative effects.

Comment: The relation between the PE and the pattern separation measure seems broad as it is not specific to any ROI and could reflect more general signal quality aspects. How do the other correlations (cue with same cue, feature with non-overlapping conjunctive stimulus) relate to the PE? Does the relative difference between overlapping pair correlation (AB-B) and unspecific correlations relate to PE?

Response: We did not find the lack of specificity between the ROIs surprising. Given our model that both the HPC and cortical inputs to the striatum influence learning, a relative increase in overlapping representations in these regions should be associated with a reduced conjunctive component of the striatal prediction error. We regret that this was not explained in the previous submission, and we now include a discussion of this point.

Nonetheless, the reviewer's concern about signal quality is important. We did two control analyses to mitigate this concern. First, the correlation between overlap in these ROIs and the striatal feature prediction error response is non-significant, $p > .2$. Signal quality fluctuations should not differentially influence conjunctive versus feature prediction error coding in the striatum.

Second, a related concern is that HPC overlap and the striatal conjunctive prediction error magnitude both vary with attention. Thus, we ran a new analysis where we included a nuisance variable that served as a proxy for individual differences in attention. This variable coded for

how well the RL model explained each subject's behavior, relative to the null model. We found that the relationship between hippocampal overlap and striatal conjunctive PE remained significant after accounting for performance in this way, $Z = -2.04$, $p = .045$. Therefore, this relationship is robust to individual differences in how well their behavior is explained by our learning model. This finding suggests that the results are not wholly due to participant engagement. However, it does not entirely exclude it as a possible contributor, and we now include consideration of this caveat in the Discussion.

Finally, the reviewer asks for additional correlations. The cue-with-same-cue correlation is what we refer to as the "identity" correlations, and these are indeed correlated with the striatal conjunctive PE, $t(31) = 2.49$, $p = .013$, $d_z = 0.45$, although this result depended on the exclusion of an outlier subject who had very high cue-with-same cue correlations (Figures S1). This finding suggests that the results were not driven by general signal quality issues, as stimulus identity and stimulus overlap showed opposing relationships to the conjunctive PE in the predicted direction. In addition, the relative difference between overlapping pairs and unspecific correlations, if we understand the reviewer correctly, is the overlap regressor. The overlap regressor captures the difference in correlations between [(AB,B), (AC, C), (AC, AB)] and [(AB, C), (AC, B), (B,C)]. We have clarified this point in the manuscript.

In order to further assess behavior in response to the concern about specificity, we conducted two new analyses. We computed an index that measures the extent to which conjunctive learning was used in each run of the subject's behavior. Specifically, we computed the likelihood of the data for each run and subject under the maximum likelihood parameters for the Value Spread model and the Conjunctive model. We computed the difference in likelihood between the models. This measure reflects the extent to which the Conjunctive model accounts for the data better than the Value Spread model. We fit a mixed-effects model of this measure with random intercepts for subjects and included a nuisance covariate that measured how well subjects learned in each run, relative to chance. This helps to ensure that the relationship is not due to fluctuations in participant engagement. Contrary to our predictions, we did not find any relationship between overlap in any of our ROIs and this measure. It is possible that variability between runs in this measure may be too large to detect fine scale relationships with behavior. We next examined whether our conjunctive learning measure was related to univariate signal magnitude, and found that runs with stronger hippocampal activity were also runs with the most conjunctive learning, $t(31) = 3.1$, $p = .01$, $d_z = 0.56$, FDR corrected. This relationship was nonsignificant in the other ROIs, all $p > 1$. Therefore, univariate signal in the hippocampus is predictive of conjunctive learning. Importantly, as we now discuss in the new Discussion, this finding is not inconsistent with a model in which attention causes both hippocampal signal to increase and conjunctive learning to increase. However, the nonsignificant finding in the other ROIs suggests that the result is not merely driven wholesale differences in engagement with the task. Rather, attention could be a latent mechanism that increases hippocampal engagement with the task in order to

improve conjunctive learning (Aly and Turk-Browne 2016). Future work is necessary to establish the role of attention. We have added discussion of this result and its relationship to attention to the manuscript.

Comment: A number of relevant studies have investigated the effects of overlap on hippocampal representations, for instance Chanalles et al., 2017, *Curr Biol.* or Favlia et al., 2017, *Nat Communications*. In addition, several studies have investigated the effects overlap of features during value learning and prediction, and their effects on hippocampal representations, Barron et al., 2013, *Nature Neurosci*. It seems remarkable that Barron et al found that HPC representations during value prediction of compound stimuli are more similar to their components than to other stimuli (i.e., the reverse of what was found here). I think these studies and their relation to the presented work should be considered. Note that I am not an author on any of the mentioned studios nor do I have close ties with the authors.

Response: We thank the reviewer for raising this important point. These papers are indeed related and deserve attention in the manuscript. The Chanelles and Favlia papers both argue that the hippocampus separates its representation of stimuli with similar outcomes over the course of learning. This result is consonant with our exploratory analysis showing that the hippocampus has more distinct representations of stimuli with similar values (i.e., similar association strengths to the target). Spurred by these papers, we examined whether this effect increased over learning. We added a regressor for epoch that was defined to be positive for comparisons between later trials in the run and negative for comparisons between earlier trials in the run. We found an interaction between the value regressor and the epoch regressor, such that the stimuli with similar associations to the target moved further apart in representational space over the course of a run. We find the opposite in mOFC, such that stimuli with similar values move closer in representational space over the course of a run. We have added this new set of results into the main text, and believe they complement our main findings and extend the impact of the work.

Barron et al found repetition suppression for compounds (AB) that were preceded by their components (A or B) relative to non-components (C) in the hippocampus. This effect would

appear to directly diverge from our finding, albeit with a different dependent measure. However, in Barron et al., this effect was eliminated if subjects were pre-exposed to the compound food (AB) about which they were constructing value estimates. Therefore, this finding only occurred when task demands explicitly required subjects to integrate across experiences. Further, the integration between singletons and conjunctions was weaker in hippocampus than in mPFC, which is broadly consistent with the difference we observed between these regions.

We note that it is difficult to fully reconcile our findings with those of Barron et al., partly due to differences in the experimental designs and dependent measures. One possibility is that the task demands for integration versus separation can influence the hippocampal code (Barron et al required integrating A and B to imagine the value of AB, whereas we require learning different outcomes for AB and B). We have added a discussion of this point to the manuscript.

Comment: Based on the Barron study and the large literature on value and state information in ventromedial prefrontal areas, it seems interesting to investigate value signals and effects of stimulus overlap in these ROIs too, especially given the caveat noted above that the current study significantly deviates from classical value based choice tasks in meaningful ways.

Response: We agree with the reviewer that the ventromedial prefrontal cortex is an interesting region to examine in this task. We included a medial orbitofrontal (mOFC) ROI. This ROI encompasses much of what is commonly referred to as vmPFC. We conducted the same analysis and found that the mOFC represents stimuli more similarly when they have a similar value (which contrasts with the findings in HPC and MTL cortical regions). We interpret this as the mOFC representing stimuli by their associated outcomes, consistent with the role of this region in outcome evaluation. We have included these new results in the manuscript.

Comment: I am not sure how the RTs were converted into likelihoods. Which distribution was assumed, and where parameters fixed? More details on the modelling would be helpful.

Response: We thank the reviewer for raising this issue, as we did not sufficiently explain our modeling. We modeled RTs using linear regression, with the difference that we jointly fit the parameters of the regression model and the parameters of the reinforcement learning model. We calculated likelihoods from the regression fits using the standard regression formula that assumes normally distributed errors (σ^2):

$$LL = n \log \left(\frac{1}{\sqrt{2\pi}\sigma} \right) - \frac{1}{2\sigma^2} \sum_{t=1}^n (rt^t - \sum_k \beta_k V_k^t)^2$$

Where n is the number of trials, the V_k^t are k different estimates of value at trial t , and the β_k are regression coefficients. There were no fixed parameters (The errors are computed empirically as the standard deviation of the residuals). The term in the summation over n is the mean-squared regression error. This is the same approach that statistical packages such as R use when

producing a log-likelihood for a linear regression model. We now include a description of this in the manuscript.

Comment: Unlike the Niv et paper, the feature RL model assumed equal weighting between features. Have the authors tried to fit feature weights too? I am almost surprised the feature RL model performs second best, given that one cannot perform well in the task when feature values are learned individually.

Response: To the best of our understanding, our model is identical to the Niv feature model but is presented in a different manner. It is customary in RL to describe feature learning as adjusting weights on unit-valued features, as described in the Niv paper. In order to emphasize the parallels with our other models, we described the feature model as learning values of the features. The “values” and “weights” are identical. We now clarify this connection to the previous literature in the manuscript.

Related to the reviewer’s second point, feature learning is indeed maladaptive in this task. However, feature learning is a prominent aspect of learning and may be the default learning strategy. We do not think subjects can “turn off” this system when it is maladaptive, but have to learn via feedback to downweight the strength of the corticostriatal synapses from sensory areas representing features. We now discuss this important point in the manuscript.

Comment: It would be helpful to briefly mention the number and nature of parameters for the different models during their description in the main text.

Response: This is a helpful suggestion, which we have implemented. The Value Spread model has three parameters (learning rate, value spread, and a regression weight on reaction time), whereas the Feature and Conjunctive models have two (value spread and regression weight).

Comment: What is d_z ? (p11 top)

Response: This refers to one of several Cohen’s d effect-size measurements. D_z is specific to paired or one-sample designs. We now include a short description the first time it is introduced.

Comment: I don’t fully follow the authors logic in the discussion when they state that “our data suggest future models that exploit the computational ...”. Is this about AI models?

Response: Our intention was to relate the current work to AI models that purport to use hippocampus-like architectures, but are really better described as having an MTL cortex-like architecture (they activate related memories in proportion to the similarity of the current stimulus). We meant to argue that incorporating into these models a module that pattern

separates similar experiences with different outcomes could improve their ability to learn. However, we agree that this paragraph was confusing and peripheral, and have removed it.

Comment: The GLMs set up to get pattern estimates that are used in the PSM seem peculiar. Why are nuisance variables for overlap and same/different target included in the analyses, when the effect overlap is part of the question? Does that come change and influence how much pattern overlap you will find? In addition, it seems unusual to have PE and value regressors, when these usually are parametric modulator assigned to a stimulus regressor.

Response: We appreciate that our description of the GLM was unclear, and thank the reviewer for drawing this to our attention. The nuisance regressors were for the mean pattern similarity of the runs, the linear and quadratic effect of time, the effect of target, and the effect of response. The response nuisance regressor accounts for similarity introduced by stimuli sharing the same outcome (either both target or both non-target). This similarity could arise from visual or cognitive factors. The target regressor accounts for the fact that similarity may not be the same for both-target and both-non-target pairs; it is an interaction between response and target. The regressors are important because these factors are unbalanced in the overlap-versus-nonoverlap critical comparison (as discussed in the comment above).

Most critically, the overlap and identity regressors were our core regressors of interest. We have clarified this in the Methods.

The value regressor was included as an exploratory effect of interest. The value regressor captures similarity between stimuli that are more similar in their target predictiveness. Unlike the target/response regressors, which are set based on the actual outcomes, the value regressor captures trial-wise fluctuations as participants learn the predictive relationships. Some theoretical models predict that stimuli in the hippocampus should be represented more similarly if they predict the same outcomes (Gluck and Meyers, 1993), whereas others predict they should be represented more distinctly (e.g., Favila and Channelles); this divergence in past arguments is why we did not have a strong hypothesis, and were interested in what this exploratory analysis would reveal. This analysis is now reported in the main text and, following the reviewer's suggestions above, we have attempted to more clearly explain it and its relationship to the literature.

The PE regressor codes trials that share similarity due to the surprisingness of the outcome. The fact that the hippocampus shows an expectancy violation response might predict increased similarity for surprising trials. However, pattern separation might orthogonalize trials with similarly surprising outcomes. This regressor was therefore another regressor of interest about which we could not make a firm a priori prediction.

The approach we used is conceptually very similar to using parametric modulators in a univariate GLM. In this case, they were parametric modulators in the GLM of pattern similarities, and the entries represented the absolute difference in value or prediction error between each of the trials. This allowed us to examine these factors in representational space, rather than in the univariate response.

References

Aly, Mariam, and Nicholas B. Turk-Browne. 2016. "Attention Promotes Episodic Encoding by Stabilizing Hippocampal Representations." *Proceedings of the National Academy of Sciences of the United States of America* 113 (4): E420–29.

Hopkins, Ramona O., Catherine E. Myers, Daphna Shohamy, Steven Grossman, and Mark Gluck. 2004. "Impaired Probabilistic Category Learning in Hypoxic Subjects with Hippocampal Damage." *Neuropsychologia* 42 (4): 524–35.

Knowlton, B. J., J. A. Mangels, and L. R. Squire. 1996. "A Neostriatal Habit Learning System in Humans - ProQuest." *Science*, January.
<http://search.proquest.com/openview/16908e7d27610a0e04267a4b47d48b12/1?pq-origsite=gscholar&cbl=1256>.

Knowlton, B. J., L. R. Squire, and M. A. Gluck. 1994. "Probabilistic Classification Learning in Amnesia." *Learning & Memory* 1 (2): 106–20.

Shohamy, Daphna, Catherine E. Myers, Ramona O. Hopkins, Jake Sage, and Mark A. Gluck. 2009. "Distinct Hippocampal and Basal Ganglia Contributions to Probabilistic Learning and Reversal." *Journal of Cognitive Neuroscience* 21 (9): 1821–33.

Reviewers' Comments:

Reviewer #1:

Remarks to the Author:

I thank the authors for a thorough response to my previous concerns. I have just two more questions, both of which are intended to clarify the contribution and impact of the authors' findings. I believe that, however either set of analyses turns out, the study stands and I will recommend publication.

Major

- The base rate model should be tested on its own, not just as a component of the Value Spread (where it is probably trading off some against Feature learning), and in particular against the feature learning model. Can the authors affirm that the feature learning model accounts for variance, or qualitative features, of the behavioral data, that are not accounted for by the base rate model? The neural data support the feature learning component, but it will be helpful to see if the behavior alone can.

- It is interesting that the model was slightly better fitting in later trials. The effect is small, but might be more clear when examining its components. Given that the key hypothesis of this paper involves the contribution of progressively learned representations, and that the paper is touching on extremely interesting questions about the transfer of control between systems across learning, I suggest these are worth a little more investigation.

For instance, this pattern may indicate that earlier trials were driven by an unmodeled process - perhaps involving exploration or sampling - or that the initialized values in the model were unrealistic (a possibility the authors refer to). More interestingly, it may indicate that the conjunctive and featural representations were more coherent, and thus behavior on their basis was more deterministic. The latter idea is consistent with the improvement in RTs over time.

In the interest of offering a specific set of suggestions, I ask that the authors do the following:

1) Plot the fit of each separate component model - Base rate, Feature, Conjunctive - across the three epochs.

2) Plot the progression of the parameters of the Value Spread model - omega, alphas, betas - across the three epochs.

Are there robust changes in these, across the epochs? Do the changes in parameter values track the neural results?

Minor

- There is some swapping back and forth between "representational" or "pattern" similarity matrices/analyses. The latter seems to be used for the acronym, the former for the expansion (though not exclusively?). Please pick one, for clarity.

- There are several spelling/grammar errors throughout. Please give the paper a thorough read-through. Just a couple examples here:

- Figure S1 caption "Accordingly, the our response-based thresholding algorithm for designating a response as fast enough to earn reward set"

- 4.3, paragraph 2 These instructionS

Reviewer #2:

Remarks to the Author:

The authors have submitted a substantive revision in which they address most concerns in a convincing manner. I appreciate the thoroughness of this revision and believe the already quite interesting initial submission has been strengthened and additional interesting results have been added.

I have only few remaining minor comments:

In relation to the main PSA analysis, I think it would be useful to updated of Figure 4 to show for each brain area the average within-cue, feature-overlap, and non-overlap correlations right next to each other (in addition to the overlap-non-overlap difference). This would make it easier for the reader to see the relative differences within each region.

I still somewhat disagree with the authors that it is "not surprising" that feature overlap in cortical areas correlates with striatal conjunction PEs. This seems to suggest that even though there is no main effect of pattern separation outside of the hippocampus, cortical "pattern separation" can also lead to more conjunctive learning. This does stand in contrast to the idea that its the role of the hippocampus alone to drive conjunctive learning, while cortical areas drive feature based learning. Although the newly presented univariate analysis suggests some degree of hippocampal specificity, I think it would be more accurate to say clearly in the manuscript (and perhaps even the abstract) that not only hippocampal representational structure influence conjunction learning signals in the BG.

Regarding the effects of overlap on the relative likelihood of the conjunctive versus value spread models (mentioned in response to one of my comments): why not use the omega parameter, instead of the likelihood differences? It would seem to be the slightly cleaner analysis to me.

I think the manuscript could use a thorough spellcheck, as I noted several issues in passing, e.g. a grammar issue in line 54, a reference to Fig 3 instead of 4 in line 328, a missing "we" in line 336

We want to thank the editor and the reviewers for their thoughtful and helpful comments on our manuscript. We were pleased that the reviewers expressed enthusiasm about both the thoroughness of the first revision and the findings in the manuscript. The reviewers proposed several additional analyses and changes to the exposition or figures. We have conducted all requested analyses and made all requested changes. We believe the manuscript has been strengthened as a result of these changes.

Reviewer #1:

I thank the authors for a thorough response to my previous concerns. I have just two more questions, both of which are intended to clarify the contribution and impact of the authors' findings. I believe that, however either set of analyses turns out, the study stands and I will recommend publication.

Major

Comment: The base rate model should be tested on its own, not just as a component of the Value Spread (where it is probably trading off some against Feature learning), and in particular against the feature learning model. Can the authors affirm that the feature learning model accounts for variance, or qualitative features, of the behavioral data, that are not accounted for by the base rate model? The neural data support the feature learning component, but it will be helpful to see if the behavior alone can.

Response: We apologize for having misinterpreted the analysis the reviewer was requesting. Following the reviewer's advice, we now compare the base-rate model, on its own, against each of the other models. We found that the base-rate model outperforms the Feature model, $p = .002$, but that there is no difference in predictive log-likelihood between the base-rate model and the Conjunctive, $p > .14$, or the Value Spread model, $p = .14$. We interpret this as indicating that both Conjunctive/Value-Spread learning and base rate effects explain a similar amount of behavioral variance. The key question is whether they are explaining different variance in behavior, or whether our learning models are simply misinterpreting base rate effects. Our combined Base Rate + Value Spread model is the best way to test this possibility because variance accounted for by base rate effects will be explained by the Base rate model. We found, in the previous revision, that regression coefficients on both Base Rate learning and Value Spread learning are significant, indicating that they both explain variance in the reaction time data. Further, parameters of the Value Spread model remain virtually unchanged after accounting for Base Rate effects. We have added these new analysis to the manuscript.

Regarding feature learning, it is important to note that the Feature RL model does not, on its own, outperform chance (Figure 2f). We had initially constructed this model as a stronger null hypothesis against which to compare Conjunctive learning. After finding that Conjunctive learning is only marginally better than Feature learning, we constructed the Hybrid model, which

outperforms Conjunctive learning. Given these results, it is most accurate to say that incorporating feature learning improves the fit of a conjunctive learner; rather than that we find evidence for feature learning in isolation. We have changed our language to make this distinction explicit.

Comment: It is interesting that the model was slightly better fitting in later trials. The effect is small, but might be more clear when examining its components. Given that the key hypothesis of this paper involves the contribution of progressively learned representations, and that the paper is touching on extremely interesting questions about the transfer of control between systems across learning, I suggest these are worth a little more investigation.

For instance, this pattern may indicate that earlier trials were driven by an unmodeled process - perhaps involving exploration or sampling - or that the initialized values in the model were unrealistic (a possibility the authors refer to). More interestingly, it may indicate that the conjunctive and featural representations were more coherent, and thus behavior on their basis was more deterministic. The latter idea is consistent with the improvement in RTs over time.

In the interest of offering a specific set of suggestions, I ask that the authors do the following:

- 1) Plot the fit of each separate component model - Base rate, Feature, Conjunctive - across the three epochs.
- 2) Plot the progression of the parameters of the Value Spread model - omega, alphas, betas - across the three epochs.

Are there robust changes in these, across the epochs? Do the changes in parameter values track the neural results?

Response: We agree with the reviewer that these are questions of theoretical importance. We implemented the reviewer's suggested analyses. We note that there might be a minor confusion

about how we split the data, as our previous results split runs into two epochs (early and late). We also showed a small effect of increased learning across the three runs. We interpret the reviewer's analyses as referring to the two epochs, because the processes the reviewer are referring to should unfold over the course of a single run. In contrast, improvement in fit across the three runs likely reflects more general effects of practice.

We refit the models with 2 learning rate parameters- one for each half- and, in the case of the Value Spread model, 2 omega parameters (which control the spread of value) for each half. We note that, even with large amounts of data, estimating individual parameters of RL models is very noisy (Ballard and McClure 2018). The inclusion of additional parameters makes fitting individual parameters even more difficult (Note that for the imaging analyses, we took the common approach of fitting a single group learning rate to regularize the parameter estimates). Indeed, the extra parameter(s) renders the predictive likelihood of all the models no different than null, Value Spread $p = .09$, all other models, $p > .1$, indicating that models with separate parameters for the two epochs overfit the training data. Nonetheless, we find that the Value Spread model still outperforms the Conjunctive, $T = 136$, $p = .028$, and Feature, $T = 147$, $p = .048$, models.

We also see a main effect of reduced learning rate late in learning, $F(1,30) = 6.11$, $p = .02$, $n^2 = .03$, but no main effect of model, $p = .12$, nor any interaction between epoch and model. In addition, there is no significant difference between the omega parameter early and late in learning, $p > .2$. We are not aware of any empirical observations of learning rate declining over the course of learning in humans, although there are unpublished observations of this phenomenon in tasks where the reward probabilities are stable (Angela Radulescu, Niv Lab, Princeton, Personal Communication). It is common to gradually reduce learning rate in machine learning applications of RL. The intuition is that it is beneficial to rapidly incorporate new information in a new environment, but as time passes, it is better to use the information you have acquired over all of your experiences. In our task, the interpretation is that subjects more heavily weight prediction errors early in learning, when they are not sure which stimuli lead to the target; but later in learning they weight prediction errors less because it is more clear which stimuli are target-predictive.

We again find an improved model fit for later trials $F(1,30) = 135, p < .001, n^2 = .2$, and a main effect of model type $F(2,63) = 9.77, p < .001, n^2 = .003$, but no interaction, $p = .12$. The fact that we find this effect when allowing for different learning rates for the two epochs suggests that changes in learning rate over time is unlikely to account for the difference in model performance.

The question remains, then, as to why the model fits better for the second epoch. Turning to the reviewer’s idea about initialization of values, we refit our models with the initialization as a free parameter and found that the best initialization point was 0 for all models (this was our original initialization). This leaves the two possibilities the reviewer mentioned: early learning could have involved more unmodeled processes, or late learning could have been over relatively more solidified feature/conjunctive representations. Due to the lack of interaction between model type and epoch on either model-fit or parameter values, it is impossible to disambiguate these possibilities. More generally, we do not find any strong evidence for transfer of control between systems over the course of learning. We have added a summary of these analyses to the Supplement.

Minor

Comment: There is some swapping back and forth between “representational” or “pattern” similarity matrices/analyses. The latter seems to be used for the acronym, the former for the expansion (though not exclusively?). Please pick one, for clarity.

Response: We thank the reviewer for catching this and we have corrected it. We now use “pattern” whenever talking about BOLD data, and retain a few uses of the word “representation” when discussing theoretical models of information coding.

Comment: There are several spelling/grammar errors throughout. Please give the paper a thorough read-through. Just a couple examples here:

- Figure S1 caption “Accordingly, the our response-based thresholding algorithm for designating a response as fast enough to earn reward set”
- 4.3, paragraph 2 These instructionS

Response: We thank the reviewer for alerting us to these spelling and grammar issues, and we have undertaken a thorough edit with this in mind.

Reviewer #2:

The authors have submitted a substantive revision in which they address most concerns in a convincing manner. I appreciate the thoroughness of this revision and believe the already quite interesting initial submission has been strengthened and additional interesting results have been added.

I have only few remaining minor comments:

Comment: In relation to the main PSA analysis, I think it would be useful to updated of Figure 4 to show for each brain area the average within-cue, feature-overlap, and non-overlap correlations right next to each other (in addition to the overlap-non-overlap difference). This would make it easier for the reader to see the relative differences within each region.

Response: We have updated Figure 4b as the reviewer suggested. We now show the feature-sharing and non-feature sharing averages along with the difference. Figure 4a shows the average within-cue correlation. We opted not to include all 3 quantities on the same panel, as the reviewer suggests, because the within-cue correlations are much higher than the between-cue correlations, and the resulting rescaling obscures the across-cue relationships. The new Figure 4b is copied here.

Comment: I still somewhat disagree with the authors that it is “not surprising” that feature overlap in cortical areas correlates with striatal conjunction PEs. This seems to suggest that even though there is no main effect of pattern separation outside of the hippocampus, cortical “pattern separation” can also lead to more conjunctive learning. This does stand in contrast to the idea that its the role of the hippocampus alone to drive conjunctive learning, while cortical areas drive feature based learning. Although the newly presented univariate analysis suggests some degree of hippocampal specificity, I think it would be more accurate to say clearly in the manuscript (and perhaps even the abstract) that not only hippocampal representational structure influence conjunction learning signals in the BG.

Response: We have followed the reviewer’s suggestion and stated both in the abstract and manuscript that representational structure in cortical ROIs is also related to striatal conjunctive learning signals. An example of our new summary of these results from Section 2.3 is: “In sum, the more the hippocampus and medial temporal lobe cortex representations overlapped for stimuli sharing features, the less striatal error signals reflected learning signals arising from a conjunctive state space.”

Comment: Regarding the effects of overlap on the relative likelihood of the conjunctive versus value spread models (mentioned in response to one of my comments): why not use the omega parameter, instead of the likelihood differences? It would seem to be the slightly cleaner analysis to me.

Response: We used the difference in likelihoods because this quantity can be robustly computed on a per-run basis, and the analysis we present is a mixed-effects analysis across runs. We did not fit the learning models and estimate omega separately for each run because we do not have enough responses per run (~20) to reliably estimate this parameter. In contrast, we can fit the data across all runs and compute the likelihood post-hoc for each run. We have clarified this point in the manuscript.

Comment: I think the manuscript could use a thorough spellcheck, as I noted several issues in passing, e.g. a grammar issue in line 54, a reference to Fig 3 instead of 4 in line 328, a missing “we” in line 336

Response: We thank the reviewer for alerting us to these spelling and grammar issues, and we have undertaken a thorough edit with this in mind.

Reviewers' Comments:

Reviewer #1:

Remarks to the Author:

The authors have satisfactorily addressed all of my comments.

Reviewer #2:

Remarks to the Author:

All my remaining comments have been addressed sufficiently and I am happy with the manuscript as it is. I recommend publication and congratulate the authors to an interesting paper.